# Structure of the DOCK2−ELMO1 complex provides insights into regulation of the auto-inhibited state

Leifu Chang[1,7], Jing Yang[1], Chang Hwa Jo[2], Andreas Boland[1,8], Ziguo Zhang[1], Stephen H. McLaughlin[1], Afnan Abu-Thuraia[3], Ryan C. Killoran[2], Matthew J. Smith[2,4,9], Jean-Francois Côté[3,5,6,9] & David Barford[1✉]

DOCK (dedicator of cytokinesis) proteins are multidomain guanine nucleotide exchange factors (GEFs) for RHO GTPases that regulate intracellular actin dynamics. DOCK proteins share catalytic (DOCK[DHR2]) and membrane-associated (DOCK[DHR1]) domains. The structurally-related DOCK1 and DOCK2 GEFs are specific for RAC, and require ELMO (engulfment and cell motility) proteins for function. The N-terminal RAS-binding domain (RBD) of ELMO (ELMO[RBD]) interacts with RHOG to modulate DOCK1/2 activity. Here, we determine the cryo-EM structures of DOCK2−ELMO1 alone, and as a ternary complex with RAC1, together with the crystal structure of a RHOG−ELMO2[RBD] complex. The binary DOCK2−ELMO1 complex adopts a closed, auto-inhibited conformation. Relief of auto-inhibition to an active, open state, due to a conformational change of the ELMO1 subunit, exposes binding sites for RAC1 on DOCK2[DHR2], and RHOG and BAI GPCRs on ELMO1. Our structure explains how up-stream effectors, including DOCK2 and ELMO1 phosphorylation, destabilise the auto-inhibited state to promote an active GEF.

[1] MRC Laboratory of Molecular Biology, Cambridge CB2 0QH, UK. [2] Institute for Research in Immunology and Cancer, Université de Montréal, Montréal, Québec H3T 1J4, Canada. [3] Montreal Institute of Clinical Research (IRCM), Montréal, QC H2W 1R7, Canada. [4] Department of Pathology and Cell Biology, Faculty of Medicine, Université de Montréal, Montréal, QC H3T 1J4, Canada. [5] Department of Biochemistry and Molecular Medicine, Faculty of Medicine, Université de Montréal, Montréal, QC H3C 3J7, Canada. [6] Department of Anatomy and Cell Biology, McGill University, Montréal, QC H3A 0C7, Canada. [7] Present address: Department of Biological Sciences, Purdue University, West Lafayette, IN 47907, USA. [8] Present address: Department of Molecular Biology, Science III, University of Geneva, Geneva, Switzerland. [9] These authors contributed equally: Matthew J. Smith, Jean-Francois Côté. ✉ email: dbarford@mrc-lmb.cam.ac.uk

RHO family small GTPases are critical regulators of cell motility, polarity, adhesion, cytoskeletal organization, proliferation, gene expression and apoptosis. These diverse functions are stimulated by the active GTP-bound state of RHO proteins that engage a diverse array of effector proteins, thereby triggering down-stream signal transduction pathways[1,2]. Conversion of these biomolecular switches to the GTP-bound state is controlled by two families of guanine nucleotide exchange factors (GEFs); the Dbl family and the DOCK family[3–6]. GEFs catalyse the release of bound GDP in exchange for GTP.

Dbl family proteins are a large group of RHO GEFs comprising a catalytic Dbl homology (DH) domain with an adjacent PH domain, within the context of functionally diverse signalling modules. The evolutionary distinct and smaller family of DOCK proteins activates CDC42 and RAC to control cell migration, morphogenesis and phagocytosis, and have been implicated as important components of tumour cell movement and invasion[7–11]. DOCK proteins exhibit high specificity, in contrast to the Dbl GEFs that often stimulate nucleotide exchange on multiple GTPases in vitro. In humans, the eleven DOCK proteins are organized into four subfamilies encoding multidomain proteins of ~2000 amino acids[7,10]. DOCK A (DOCK1/DOCK180, DOCK2 and DOCK5) and DOCK B (DOCK3 and DOCK4) subfamilies activate RAC, whereas the DOCK D subfamily (DOCK9/Zizimin1, DOCK10 and DOCK11) activates CDC42[12–15], with only DOCK10 also activating RAC[16]. DOCK6 and DOCK7 of the DOCK C (DOCK6, DOCK7 and DOCK8) subfamily are dual specificity GEFs with activity towards both RAC and CDC42 in vivo[17–19], a finding recently confirmed for DOCK7 in vitro[20], whereas DOCK8 was shown to be a CDC42 GEF[21].

All DOCK proteins contain a catalytic DHR2 domain of ~450 residues situated within their C-terminal region (DOCK^DHR2)[13,15] (Fig. 1a). The DHR2 domain is divergent across the family, with the DHR2 domains of DOCK1 (RAC specific) and DOCK9 (CDC42 specific) sharing only 22% sequence identity[13,15]. A second region of common similarity is the ~200 residue DHR1 domain located towards the N-terminus (DOCK^DHR1) (Fig. 1a). The DHR1 domain of DOCK proteins adopts a C2-like

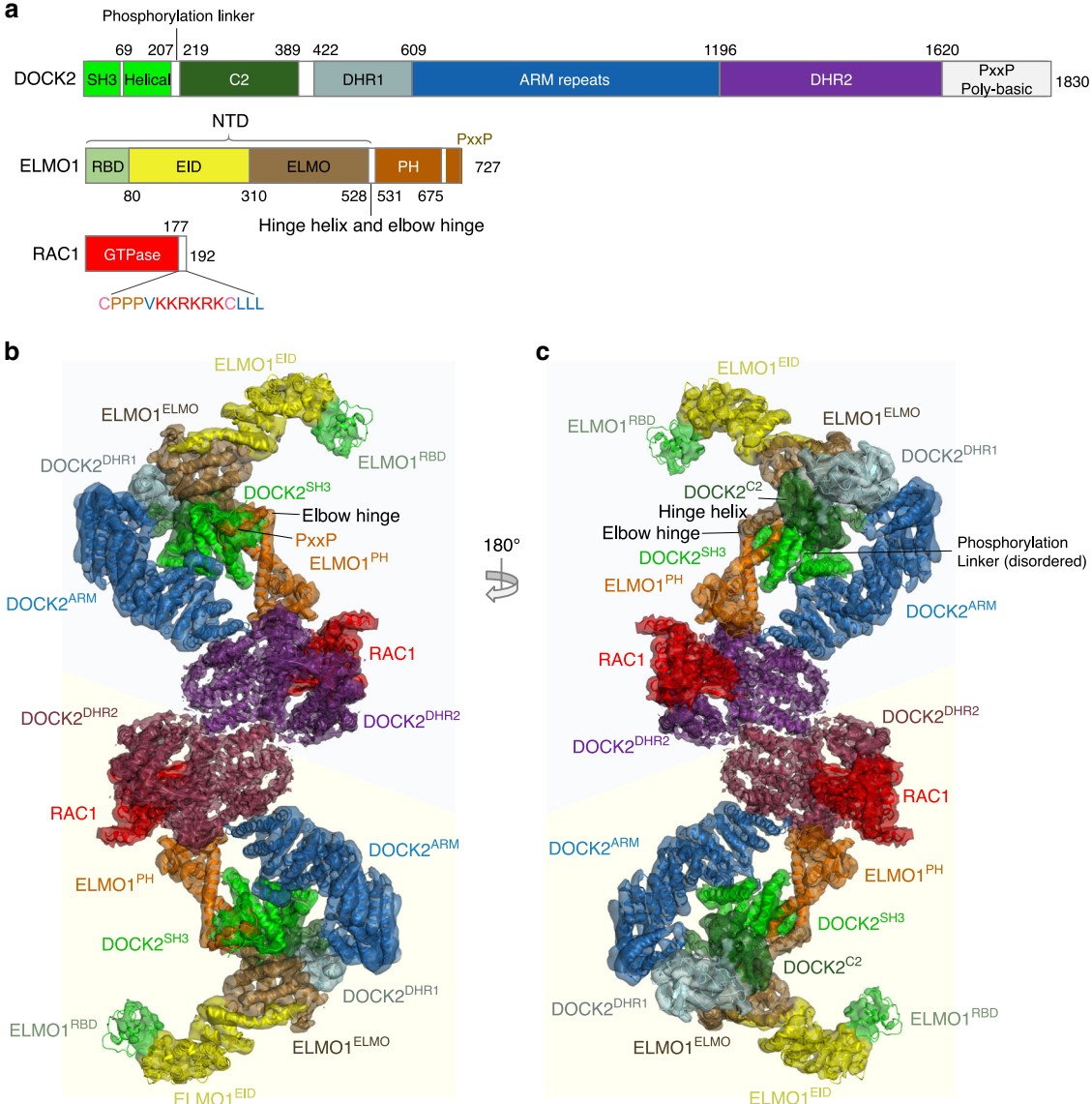

**Fig. 1 Overall structure of the DOCK2−ELMO1−RAC1 complex. a** Schematic of the domain structures of DOCK2, ELMO1 and RAC1. **b, c** Two views of the cryo-EM map of the DOCK2−ELMO1−RAC1 complex with each domain colour-coded. Ribbon representation of the structural models were placed in the cryo-EM density map. The two DOCK2−ELMO1−RAC1 protomers of the dimeric complex are indicated by light blue and light yellow backgrounds. The open-conformation of the DOCK2−ELMO1−RAC1 complex is shown.

architecture and interacts with PI(3,4,5)$P_3$ to mediate signalling and membrane localization[22,23]. DOCK proteins locate primarily to the cytosol, but recruitment to the cell membrane is critical for their roles in cytoskeleton reorganization[24–26]. DOCK A and B subfamilies incorporate an N-terminal SH3 domain and an extreme C-terminal poly-proline sequence (Fig. 1a). In contrast, the DOCK D subfamily incorporates an N-terminal PH domain, whereas the DOCK C subfamily lacks recognizable SH3 or PH domains.

Stimulation of RAC-induced cytoskeletal reorganization by the DOCK A and B subfamilies is dependent on interactions with ELMO proteins[26–29]. The SH3 domain and neighbouring α-helical region of the DOCK A and B subfamilies mediate their interactions with ELMO subunits[30,31]. ELMO proteins comprise an N-terminal domain (ELMO$^{NTD}$), a non-canonical PH domain (ELMO$^{PH}$), and a C-terminal poly-proline sequence (Fig. 1a). ELMO$^{NTD}$ itself is composed of an N-terminal RAS-binding domain (ELMO$^{RBD}$), EID and ELMO domains (ELMO$^{EID}$ and ELMO$^{ELMO}$) (Fig. 1a). Disruption of DOCK1−ELMO interactions abrogated DOCK1's ability to promote RAC-dependent cytoskeletal changes. Compared with DOCK1 alone, the DOCK1−ELMO complex binds to nucleotide-free RAC more efficiently and shows higher GEF activity[12,26–28].

Among DOCK proteins, DOCK2 plays multiple roles in regulating immune responses[32]. It is a haematopoietic cell protein which functions downstream of chemokine receptors to control actin reorganization, thus, regulating lymphocyte activation, migration, and morphology[32,33]. In T cells DOCK2 acts downstream of the T cell receptor, and recently DOCK2 was implicated in mediating signalling from the FLT3 protein tyrosine kinase and identified as a leukaemia drug target[32,33]. Through binding to the C-terminal polybasic region of DOCK2, phosphatidic acid stabilizes the recruitment of DOCK2 to the cell membrane to mediate neutrophil chemotaxis[34].

In vivo, RHOG−GTP interacts with ELMO1$^{RBD}$, an interaction that is required for DOCK1-mediated RAC1 activation[35,36]. Furthermore, the Arl family GTPase Arl4A promotes actin cytoskeleton remodelling, dependent on ELMO1$^{RBD}$, acting as a membrane localization signal for ELMO1[37]. The adhesion-type BAI (brain-specific angiogenesis inhibitor) subfamily of GPCRs are up-stream regulators of the DOCK−ELMO−RAC signalling module, controlling engulfment and degradation of apoptotic cells by phagocytes[38], and promoting myoblast fusion[39,40]. BAI receptors bind ELMO through their conserved C-terminal region[38,39], proposed to become exposed for ELMO-binding upon receptor activation[41]. A recent crystal structure of ELMO2$^{NTD}$ (lacking ELMO2$^{PH}$) revealed that this region of BAI1 forms an α-helical segment that interacts with ELMO2$^{EID}$[42].

The molecular basis for the GEF activity of DOCK proteins was elucidated from crystal structures of DOCK2$^{DHR2}$ and DOCK9$^{DHR2}$ in complex with their cognate GTPases RAC and CDC42, respectively[31,43,44], showing that GDP release and discharge of the activated GTP-bound CDC42 and RAC are catalysed by a universal invariant valine residue that functions as a nucleotide sensor[44]. However, the regulation of DOCK proteins and the signal transduction events responsible for their activation are poorly understood. It is not yet known how ELMO proteins regulate DOCK-mediated GEF activity or how effector proteins such as RHOG and BAI receptors regulate DOCK−ELMO GEF activity. To understand the molecular architecture and mechanism of DOCK−ELMO GEFs, we have used cryo-electron microscopy to determine structures of DOCK2−ELMO1 alone and as a ternary complex with nucleotide-free RAC1, and combined this with a crystal structure of ELMO2$^{RBD}$ in complex with RHOG. The binary DOCK2−ELMO1 complex adopts a closed, auto-inhibited

conformation, whereas in the DOCK2−ELMO1−RAC1 ternary complex, DOCK2−ELMO1 adopts an open, active conformation through a conformational change of the ELMO1 subunit. This exposes binding sites for RAC1 on DOCK2$^{DHR2}$, with ELMO1 also directly contacting RAC1 to promote binding to DOCK2−ELMO1. Binding sites for RHOG and BAI GPCRs on ELMO1 also become accessible in the active, open conformation. Our study suggests a model for how upstream regulators control DOCK2 activation through a conformational change of ELMO1, thereby relieving autoinhibition.

## Results

**Cryo-EM structure determination of DOCK2−ELMO1−RAC1.** We prepared the human DOCK2−ELMO1 complex using the baculovirus-insect cell expression system (Supplementary Fig. 1a). The molecular weight of the complex of ~600 kDa estimated by size exclusion chromatography (Supplementary Fig. 1b) indicated the formation of a tetramer formed of two DOCK2−ELMO1 protomers. This is consistent with the observation that DOCK2 homo-dimerization is required for DOCK2-mediated RAC activation in vivo and for lymphocyte migration[34], and with previous DHR2 domain crystal structures[31,43,44]. Purified DOCK2−ELMO1 forms a stable complex with nucleotide-free RAC1 (Supplementary Fig. 1a, b), which was further stabilized by crosslinking to avoid disassociation during cryo-EM grid preparation. We then analysed the structure of the DOCK2−ELMO1−RAC1 complex by single particle cryo-EM. Examination of raw particle images and reference-free 2D class averages indicated a considerable degree of structural flexibility (Supplementary Fig. 1c–e), limiting the resolution of the overall 3D reconstruction. To improve the EM density map quality, we generated two particle sets, each with one protomer subtracted from the original images. Refinement of the combined particles resulted in an EM density map with an overall resolution of 4.1 Å (Supplementary Fig. 2a, b). The DHR2 domain was the best-resolved feature, with other domains resolved at lower resolution (Supplementary Fig. 2c). To improve the resolution of individual structural segments, we performed focussed 3D classification on individual rigid modules, followed by focussed refinement, and obtained reconstructed maps of the (i) DOCK2$^{DHR2}$−RAC1 catalytic module at 3.8 Å resolution, (ii) DOCK2$^{ARM}$ domain at 4.2 Å, (iii) DOCK2$^{SH3}$−ELMO1$^{PH}$ at 4.1 Å, (iv) DOCK2$^{DHR1}$−DOCK2$^{C2}$ at 4.6 Å and (v) ELMO1$^{NTD}$ at 6.2 Å resolution (Supplementary Fig. 2b, c and Supplementary Tables 1–3). From the whole dataset, we selected ~2% of the particles to reconstruct a map representing the entire complex at a resolution of 7.8 Å with twofold symmetry (Supplementary Fig. 3). Reconstruction without imposed symmetry resulted in a very similar map, although at lower resolution of 9.1 Å (Supplementary Fig. 3e). Figure 1b, c shows a composite map consisting of the individual maps of each domain together with a fitted model.

**Structure of the DOCK2−ELMO1−RAC1 ternary complex.** The ternary DOCK2−ELMO1−RAC1 dimeric complex adopts an elongated "S"-like shape with pseudo twofold symmetry, measuring 320 Å and 65 Å in the longest and shortest dimensions, respectively (Figs. 1b, c and 2a). Both the DOCK2 and ELMO1 subunits are assembled from a series of modular domains (Fig. 1a). In the complex, these two subunits are arranged in a roughly parallel manner, but because DOCK2 adopts a hook-like structure (Fig. 2b), its N-terminal DOCK2$^{SH3}$ domain is positioned to interact with the C-terminal PxxP motif of ELMO1 (Figs. 1b and 2a). Located at the centre of DOCK2−ELMO1−RAC1 is DOCK2$^{DHR2}$, dimerized through its A lobe, and associated with nucleotide-free RAC1 through its B and C lobes (Figs. 1b and 2a, c). Side-chains are visible

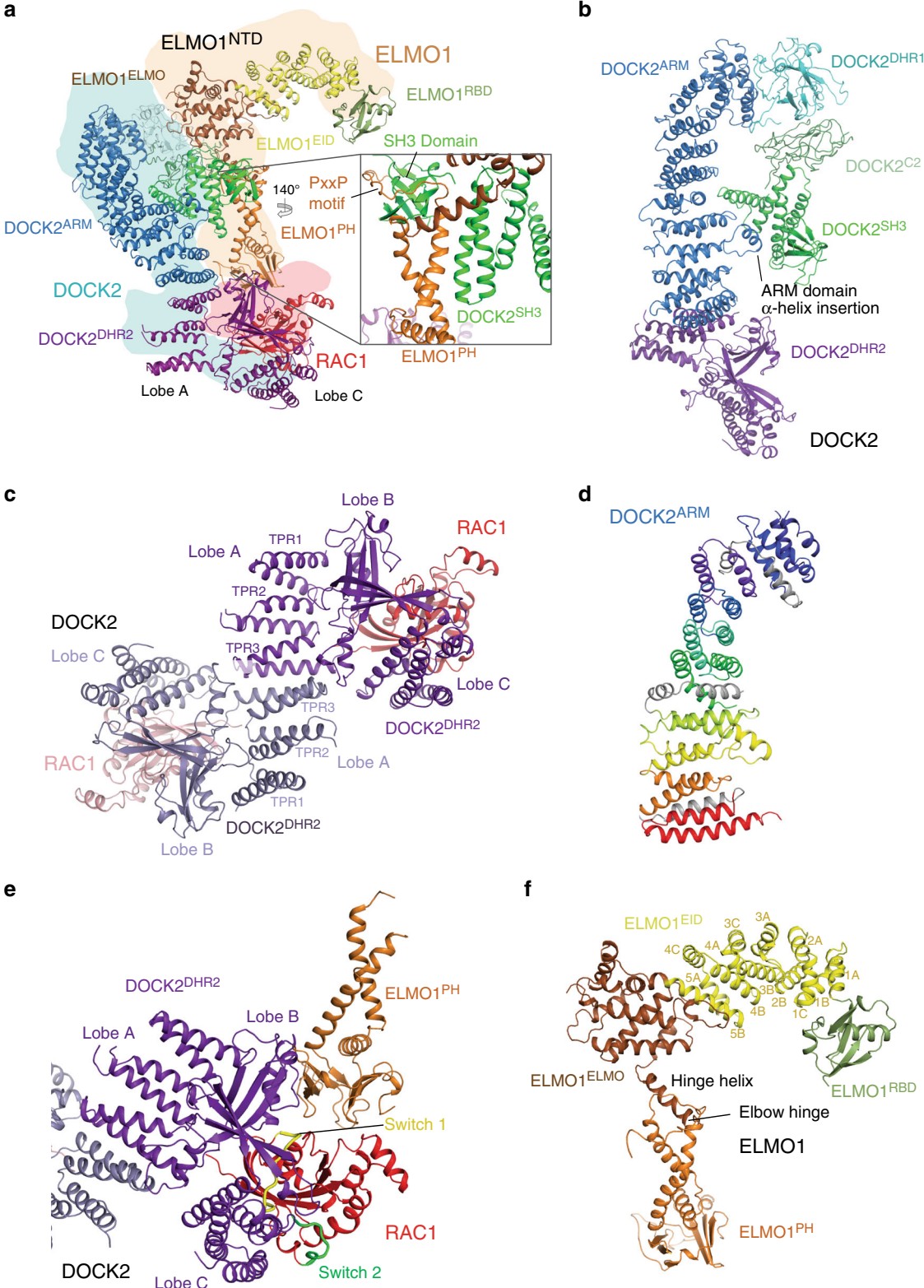

**Fig. 2 Structure of the DOCK2–ELMO1–RAC1 protomer. a** Ribbon representation of the structural assembly of the DOCK2–ELMO1 protomer. DOCK2, ELMO1 and RAC1 are highlighted in cyan, yellow-orange and red backgrounds, respectively. Insert shows the major stable interface between DOCK2 and ELMO1 involving the six α-helical bundle. **b** Ribbon representation of DOCK2. Domains are labelled and colour-coded according to the domain scheme in Fig. 1a. **c** Close-up view of the dimer interface formed by lobe A of the DHR2 domains of each DOCK2 monomer (shown in purple and grey-blue). Lobe A, B and C of the DOCK2 DHR2 domains are labelled. **d** DOCK2$^{ARM}$ is rainbow-coloured from blue (N-terminus) to red (C-terminus). **e** Close-up view of the interaction interface between DOCK2$^{DHR2}$, ELMO1$^{PH}$ domain and RAC1. The switch 1 and switch 2 loops of RAC1 are coloured in yellow and green, respectively. **f** Ribbon representation of ELMO1 colour-coded according to Fig. 1a.

in this region (Supplementary Fig. 4a), allowing atomic model building. The polybasic region C-terminal of DOCK2$^{DHR2}$ (Fig. 1a) is disordered. The region immediately N-terminal of DOCK2$^{DHR2}$, and connected to the lobe A TPR repeats of DOCK2$^{DHR2}$, is a right-handed α-solenoid domain. This is formed of 28 α-helices composed of a mixture of ARM and HEAT repeats, termed DOCK2$^{ARM}$ (Figs. 1 and 2b, d). A DALI search[45] revealed the vacuolar protein 8, an ARM repeat protein[46], as the closest match (Z score of 11.8, RMSD of 4.4 Å based on 290/568 aligned residues). A short α-helix inserted within the ARM domain, projects out from the main α-solenoid (Fig. 2b). This α-helical insert provides a major site of interaction between DOCK2$^{ARM}$ and DOCK2$^{SH3}$ (Fig. 2b).

Connected to DOCK2$^{ARM}$ is a globular density feature composed of two closely packed domains. A homology model of DOCK2$^{DHR1}$ based on the crystal structure of DOCK1$^{DHR1}$[23] was confidently fitted to the domain closest to DOCK2$^{ARM}$, consistent with the connectivity of DOCK2$^{DHR1}$ and DOCK2$^{ARM}$ in the protein sequence (Figs. 1a, c and 2b). The other globular domain (residues 219–389), located between DOCK2$^{DHR1}$ and DOCK2$^{SH3}$ is mainly composed of β-strands, consistent with secondary structure predictions. A β-sandwich structure composed of two β-sheets each with four anti-parallel β-strands, predicted using Rosetta[47], was fitted to this β-sheet-like cryo-EM density (Figs. 1b, c and 2b). A DALI search[45] revealed this β-sandwich domain to be a C2-domain, with the highest structural similarity to the C2A domain of Otoferlin[48] (Z score of 4.8). Thus, DOCK2$^{DHR1}$[23] and DOCK2$^{C2}$ share related C2-fold architectures. DOCK2$^{C2}$ is connected to the N-terminal DOCK2$^{SH3}$ and its adjacent α-helical domain by a flexible region found to be phosphorylated in multiple mass spectrometry studies (PhosphoSitePlus). We termed this the phosphorylation linker (Fig. 1a, c).

The structure of the DOCK2$^{DHR2}$−RAC1 module in the context of the entire DOCK2−ELMO1−RAC1 complex (Fig. 2e) is virtually identical to the crystal structure of the isolated DOCK2$^{DHR2}$−RAC1 complex[43], with a RMSD between them of less than 1 Å. DOCK2$^{DHR2}$ forms the dimer interface of the two DOCK2−ELMO1−RAC1 protomers (Fig. 2c), similar to other DOCK proteins revealed by structures of their DHR2 domains that are dimers, for example DOCK9$^{DHR2}$ (Ref. [44]), and as we previously published for DOCK2$^{DHR2}$ (Ref. [43]). Lobe A contains three TPR motifs with TPR3 forming the main DOCK2 dimerization interface. We produced a monomeric complex of DOCK2−ELMO1 by mutating the dimerization interface of lobe A of DOCK2$^{DHR2}$ (Supplementary Fig. 5a–c). Cryo-EM examination showed that monomers are less stable. About 90% of particles were disassembled, with the remaining particles reconstructed to generate a low-resolution map (Supplementary Fig. 5d, e). This result indicates that DOCK2 dimerization stabilizes the complex. Although there are no clear functional explanations for why DOCK proteins dimerize, one possibility would be to enable cooperativity between the two catalytic sites of the DOCK dimer.

ELMO1 comprises ELMO1$^{NTD}$, ELMO1$^{PH}$ and C-terminal PxxP motif (Fig. 1a). ELMO1$^{NTD}$ and ELMO1$^{PH}$ are connected by a single α-helix termed the hinge helix (Fig. 2f). A flexible elbow hinge at the C-terminus of the hinge helix allows mobility of ELMO1$^{NTD}$ to adopt open- and closed-conformations (Fig. 3a and Supplementary Fig. 3a, b). 3D classification showed that in the DOCK2−ELMO1−RAC1 ternary complex, ELMO1$^{NTD}$ is predominantly in the open-conformation (Supplementary Fig. 3a–d). In this conformation, ELMO1$^{NTD}$ is situated above DOCK2$^{SH3}$ (Figs. 1b and 3a – left panel), whereas in the closed-conformation, ELMO1$^{NTD}$ is poorly ordered, with its tip projected towards DOCK2$^{DHR2}$ (Fig. 3a – right panel). In the ternary DOCK2−ELMO1−RAC1 complex (open-conformation),

ELMO1 adopts an elongated, gently curved shape (Fig. 2f). We fitted the crystal structure of residues 1–520 of ELMO2 (ELMO2$^{NTD}$, PDB:6IDX)[42] (74% sequence identity to ELMO1$^{NTD}$) to cryo-EM density assigned to ELMO1$^{NTD}$. ELMO2$^{NTD}$ superimposes closely with ELMO1$^{NTD}$ within the context of the DOCK2−ELMO1−RAC1 complex. ELMO1$^{EID}$ of ELMO1$^{NTD}$ is composed of five pairs of anti-parallel α-helices (Fig. 2f). Helix 5B is characterized by its length and protrusion into the ELMO domain of ELMO1$^{NTD}$ (Fig. 2f). The EID and ELMO domains together are common to six human proteins; ELMO1, ELMO2, ELMO3, ELMOD1, ELMOD2, and ELMOD3. ELMO1$^{RBD}$ at the N-terminus of ELMO1, adopts a ubiquitin-like fold that is exposed to solvent and accessible for interactions with RHOG[35] and Arl4[37] (Figs. 1b, c and 2a).

The major and stable DOCK2−ELMO1 interface is generated by the PH domain and C-terminal PxxP motif of ELMO1 associating with the N-terminal SH3 domain and adjacent α-helical segment of DOCK2 (Figs. 1a, b and 2a). This is essentially identical to the crystal structure of the isolated DOCK2$^{SH3}$−ELMO1$^{PH}$ assembly[31]. This interface comprises two segments. First, a six-α-helical bundle produced by the accretion of an α-helical segment adjacent to DOCK2$^{SH3}$ with the two α-helices that flank ELMO1$^{PH}$[31] (Fig. 2a, insert). Second, the PxxP motif immediately C-terminal to ELMO1$^{PH}$ engages DOCK2$^{SH3}$, as also previously defined[31] (Figs. 1b and 2a – insert). In the cryo-EM structure of the DOCK2−ELMO1−RAC1 complex (as for the DOCK2−ELMO1 binary complex described below), there are no direct contacts between DOCK2$^{SH3}$ and DOCK2$^{DHR2}$, although such interactions may exist in DOCK2 alone as suggested by biochemical data for DOCK1 and DOCK2[31,49].

In an interaction that is important for DOCK2 GEF activity[12,50–53], ELMO1$^{PH}$ is positioned in a groove created between lobe B of DOCK2$^{DHR2}$ and the nucleotide-free RAC1 (Figs. 1b, c, and 2a, e). This is consistent with an earlier study showing that DOCK1 and ELMO1$^{PH}$ interact directly[30]. In contrast to the PH domains of Dbl family GEFs such as Sos1, the non-canonical ELMO1$^{PH}$ lacks critical phosphoinositide-binding residues, and is not involved in membrane attachment[30]. However, it is critical for optimal DOCK GEF activity as shown in the nucleotide exchange assay (Supplementary Fig. 6a), consistent with previous observations both in vitro and in vivo[12,50–53]. A possible mechanism is that ELMO1 stabilizes DOCK2 by interacting with DOCK2$^{SH3}$ through ELMO1$^{PH}$ (Fig. 2a). This is consistent with our observation that the purified DOCK2 protein alone (without ELMO1) elutes from a size-exclusion column as a broad extended peak (Supplementary Fig. 6b), and that in negative stain EM micrographs it appears highly heterogeneous with a tendency to aggregate (Supplementary Fig. 6c). In addition, our structure suggests that ELMO1$^{PH}$ is directly involved in DOCK2 GEF activity by interacting simultaneously with DOCK2$^{DHR2}$ and nucleotide-free RAC1, thereby stabilizing DOCK2$^{DHR2}$–RAC1 interactions (Fig. 2e).

**DOCK2−ELMO1 structure shows conformational change of ELMO1.** To understand potential regulatory mechanisms that underlie DOCK2−ELMO1 functions, we also analysed the structure of the DOCK2−ELMO1 binary complex by cryo-EM (Supplementary Fig. 7 and Supplementary Tables 1 and 2). 3D classification showed that in the DOCK2−ELMO1 binary complex, ELMO1$^{NTD}$ adopts two conformations: the open and closed-conformations (Fig. 3b and Supplementary Fig. 7). In ~12% of DOCK2−ELMO1 particles, ELMO1$^{NTD}$ adopts the open-conformation, resembling that of the DOCK2−ELMO1

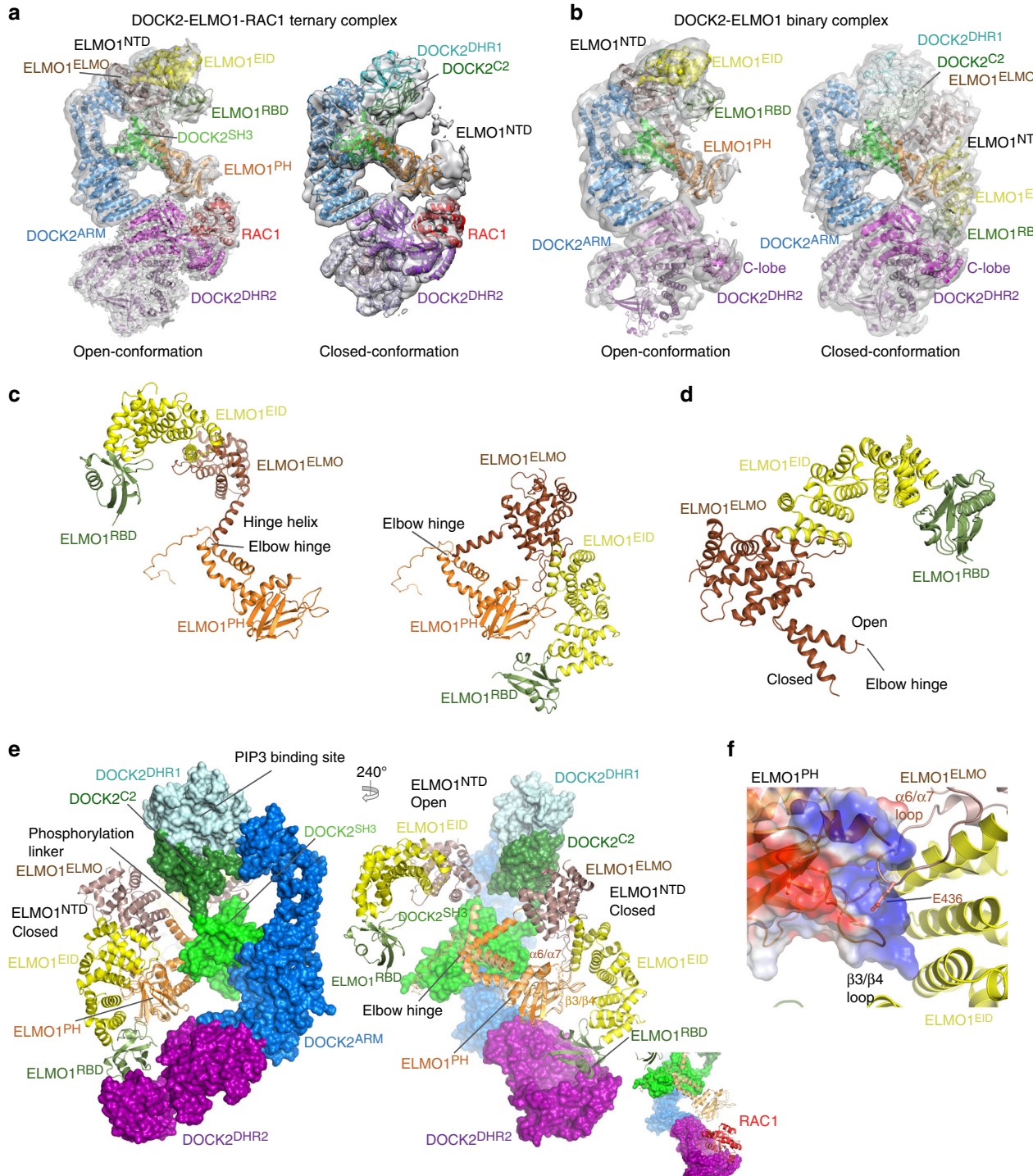

ternary complex with RAC1 (Fig. 3a, b – left panels and Supplementary Fig. 7d). However, in the majority (~80%) of DOCK2−ELMO1 binary complex particles, ELMO1$^{NTD}$ adopts a closed-conformation (Fig. 3b – right panel). This conformation differs slightly from the closed-conformation of the ternary DOCK2−ELMO1−RAC1 complex. In the DOCK2−ELMO1 binary complex, ELMO1$^{NTD}$ is more ordered through interactions with DOCK2$^{DHR2}$, whereas ELMO1$^{NTD}$ adopts multiple conformations in the DOCK2−ELMO1−RAC1 complex. Interconversion between the open DOCK2−ELMO1−RAC1 ternary complex and the closed DOCK2−ELMO1 binary complex

involves ELMO1$^{NTD}$ undergoing a rigid-body rotation of ~120°, centred on the elbow hinge that connects ELMO1$^{NTD}$ and ELMO1$^{PH}$ (Fig. 3c–e and Supplementary Movie 1). The closed-conformation is stabilized by ELMO1 intra-domain contacts. Specifically, the long flexible loop, inserted between α6 and α7 of ELMO1$^{ELMO}$, together with the neighbouring ELMO1$^{EID}$, form intimate contacts with ELMO1$^{PH}$ (Fig. 3e – right panel and Fig. 3f). The tip of the 16-residue α6/α7 loop of ELMO1$^{ELMO}$ contacts the positively charged surface of ELMO1$^{PH}$, with the invariant Glu436 of the ELMO1$^{ELMO}$ α6/α7 loop inserting into a groove created by the basic β3/β4 loop of ELMO1$^{PH}$. This β3/β4 loop in

**Fig. 3 Mechanism of DOCK2-ELMO1 activation. a** Comparison of the open- and closed-conformations of the DOCK2–ELMO1–RAC1 ternary complex. Left panel: cryo-EM density map of the complex adopting the open-conformation, with model coordinates placed into the map. Right panel shows the closed-conformation. ELMO1$^{NTD}$ is less well defined. **b** Comparison of the open- and closed-conformations of ELMO1 in the binary DOCK2–ELMO1 complex. A ribbon representation of the model has been placed in the cryo-EM density maps. **c** Ribbon representation of ELMO1 shown in the open-conformation of the ternary complex (left) and the closed-conformation of the binary complex (right). Domains are labelled and colour-coded according to Fig. 1a. **d** ELMO1$^{NTD}$ of the ternary and binary states are superimposed. This illustrates that the conformational change between the two states is confined to a rotation about the hinge elbow that connects ELMO1$^{ELMO}$ and ELMO1$^{PH}$. For clarity ELMO1$^{PH}$ is omitted from Figure. **e** Two views showing superpositions of the open DOCK2–ELMO1–RAC1 ternary complex and the closed DOCK2–ELMO1 binary complex. In both views, DOCK2 of the ternary complex is shown as a surface representation, whereas ELMO1 is shown as a ribbon representation. This highlights the conformational rearrangement due to rotation of ELMO1 about the elbow hinge. In the closed-conformation of ELMO1, ELMO1$^{RBD}$ contacts DOCK2$^{DHR2}$ at the RAC1-binding site, inhibiting interactions of RAC1 with DOCK2$^{DHR2}$ (insert: ternary DOCK2–ELMO1–RAC1 complex). In the left view, the DOCK2 phosphorylation linker is labelled. In the closed-conformation of DOCK2–ELMO1, this region is in close proximity to ELMO1$^{ELMO}$ and the adjacent hinge helix, suggesting that phosphorylation at this site would favour the open active conformation. ELMO1$^{PH}$ of the binary and ternary complexes are coloured orange and light orange, respectively. **f** Details of the interface between ELMO1$^{PH}$ (displayed as an electrostatic potential surface) with ELMO1$^{NTD}$, specifically the ELMO1$^{ELMO}$ and ELMO1$^{EID}$ domains of the DOCK2–ELMO1 binary complex.

turn projects out from ELMO1$^{PH}$ to insert into the acidic concave groove formed by the ARM repeats of ELMO1$^{EID}$ (Fig. 3f).

In the closed-conformation of the DOCK2−ELMO1 binary complex, ELMO1$^{NTD}$ also forms extensive interactions with DOCK2 (Fig. 3b, e). Importantly, ELMO1$^{RBD}$ is positioned to directly contact the RAC1-binding site of DOCK2$^{DHR2}$. In addition, ELMO1$^{ELMO}$ contacts DOCK2$^{C2}$ (Fig. 3b, e), in contrast with the open-conformation of ELMO1$^{NTD}$ which forms no contacts with DOCK2 (Fig. 1b, c). Superimposition of binary and ternary structures (on DOCK2$^{DHR2}$) indicates that ELMO1$^{RBD}$ in the closed-conformation of the DOCK2−ELMO1 binary complex, and RAC1 in the DOCK2−ELMO1−RAC1 ternary complex, overlap (Fig. 3e). Thus, in the binary state with ELMO1$^{NTD}$ in the closed-conformation, ELMO1$^{RBD}$ sterically occludes RAC1 engagement, suggesting an auto-inhibited conformation (Fig. 3e).

Another conformational difference between the binary and ternary complexes is located in lobe C of DOCK2$^{DHR2}$. In the binary complex, lobe C is rotated away from lobes A and B of DOCK2$^{DHR2}$, with a maximum shift of ~10 Å. This allows ELMO1$^{RBD}$ to engage DOCK2$^{DHR2}$ (Supplementary Fig. 4b and Supplementary Movie 1). In this conformation, as suggested by the weaker, less well-defined EM density, DOCK2$^{DHR2}$ is more flexible than in the ternary complex. This change is mainly due to the absence of RAC1 because lobe C adopts similar conformations in both the open- and closed-conformations of the DOCK2−ELMO1 binary complex (Fig. 3b).

**Crystal structure of RHOG−GMP-PNP complexed with ELMO2$^{RBD}$.** Binding of RHOG to ELMO may regulate RAC1 activation by spatially restricting DOCK-ELMO complexes in cells, or through direct molecular regulation of GEF activity. Whether ELMO$^{RBD}$ uses a similar binding mode as RBDs found in RAS GTPase effectors is unknown. To answer these questions, we determined a crystal structure of activated RHOG complexed with the RBD domain of ELMO2 (ELMO2$^{RBD}$). The crystals diffracted to 2.4 Å resolution with a single molecule in the asymmetric unit (Fig. 4a and Supplementary Table 4). The RHOG structure consists of six β-strands and six α-helices and is similar in structure to other RHO family small GTPases. Characteristically, RHOG nucleotide binding is coordinated by two key regions: switch 1 and switch 2 loops which bind the terminal γ-phosphate of GTP and a divalent magnesium ion (Fig. 4a, c). ELMO2$^{RBD}$ demonstrates a classical ubiquitin-like fold consisting of four β-strands and two α-helices, but interacts with RHOG at a non-canonical binding interface (Fig. 4b, c).

RAS is archetypally complexed with effector RBDs through an intermolecular, anti-parallel β-sheet consisting of β2 and β3

(switch 1) of the GTPase and β1, β2 and α1 of the RBD[54]. In contrast, the interface between RHOG and ELMO2$^{RBD}$ comprises both switch 1 and switch 2 regions of the GTPase (Fig. 4b, c). Major interactions include a hydrogen bond between ELMO2 Lys9 and the main chain at RHOG Phe37, as well as a salt bridge between Arg66 of RHOG and Glu13 of the RBD (Fig. 4c). In addition, there are extensive hydrophobic interactions between side chains of five RHOG residues (Val36, Phe37, Tyr64, Leu67 and Leu70) and four ELMO2 residues (Ala11, Ala19, Leu21 and Ile74) (Fig. 4c). Each of these amino acids are highly evolutionarily conserved in vertebrates (Supplementary Fig. 8). There is no direct ortholog of RHOG in simple organisms, the most homologous being RAC orthologs, but all key binding residues are conserved in these GTPases. Interestingly, the side chain of Arg66 is surface exposed in every available structure of a RAC GTPase. This residue is nonetheless invariant through evolution, suggesting a potential role in mediating protein interactions, as observed in our structure (Fig. 4c). For ELMO, many orthologs in less complex organisms have substitutions at Glu13, but key hydrophobic residues and particularly Lys9 are well conserved. Thus, the crystal structure of RHOG complexed with ELMO2 elucidates a novel GTPase-RBD binding interface dependent on several key residues that are highly conserved evolutionarily.

**Mutations disrupt RHOG−ELMO2$^{RBD}$ interactions.** To validate the RHOG−ELMO2$^{RBD}$ crystal structure and identify mutations that can disrupt formation of the RHOG−ELMO2$^{RBD}$ complex, we performed binding assays using isothermal titration calorimetry (ITC). We first determined the nucleotide dependency of this interaction by performing ITC with GDP- or GMPPNP-loaded RHOG and the ELMO2$^{RBD}$. As shown in Fig. 5a, GDP-bound RHOG does not interact with ELMO2$^{RBD}$, while GTP analogue-bound RHOG interacts with an affinity of 7.8 μM. This result is consistent with the paradigm of small GTPase signalling, and with our structural data revealing the interaction involves the nucleotide-sensitive switch regions.

We were unable to crystallize RHOG in the GDP bound state, but a model based on available RHO family GDP-bound structures demonstrates that RHOG switch 1 would move significantly outward upon hydrolysis of GTP (Fig. 4b and Supplementary Fig. 9a). This would likely disrupt hydrogen bonding between ELMO2 Lys9 and the backbone of RHOG (Fig. 4c shows the position of this residue at the binding interface). Indeed, ITC analysis of an ELMO2 Lys9 to Ala mutant shows the importance of this Lys side chain, as binding to RHOG is completely abrogated (Fig. 5b). Our crystal structure revealed this is a backbone interaction with Phe37 of RHOG switch 1 rather than a salt bridge with RHOG Asp38 (Fig. 4c). Supporting

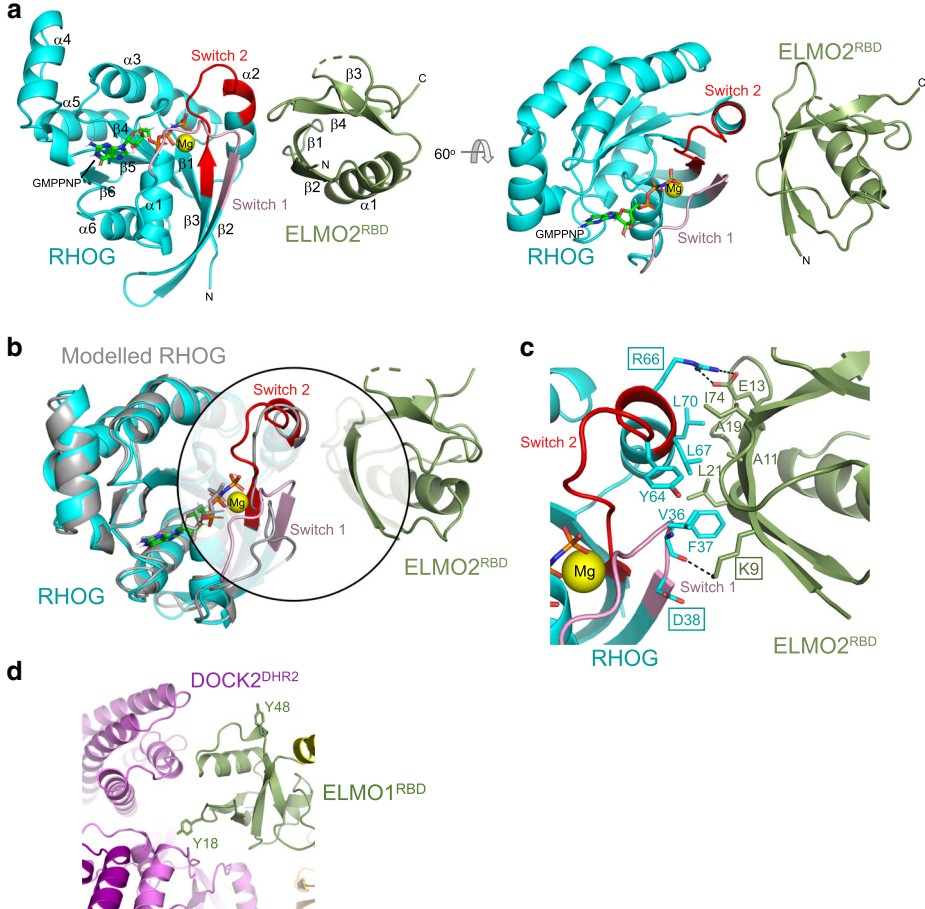

**Fig. 4 Crystal structure of the RHOG–ELMO2$^{RBD}$ complex. a** Structure of RHOG complexed with ELMO2$^{RBD}$. The switch 1 (pink) and switch 2 (red) regions of RHOG are highlighted. GMPPNP is shown in stick form and the bound Mg$^{2+}$ ion (yellow). **b** Superimposition of the RHOG–ELMO2$^{RBD}$ structure with that of the modelled GDP-bound RHOG (grey, based on RHOA; PDBid 1FTN). The switch 1, switch 2 and binding interface with ELMO2$^{RBD}$ are highlighted in the circle. The switch 1 region of inactive RHOA is noticeably shifted outward. **c** Detailed view of the RHOG–ELMO2$^{RBD}$ binding interface. Residues involved in the interaction are labelled and shown with side chains in stick form. **d** In the closed-inactive conformation of DOCK2–ELMO1, Tyr18 contacts DOCK2$^{DHR2}$. Phosphorylation of Tyr18 by TAM kinases would destabilize DOCK2$^{DHR2}$–ELMO1$^{RBD}$ interactions, thereby stimulating DOCK2 GEF activity.

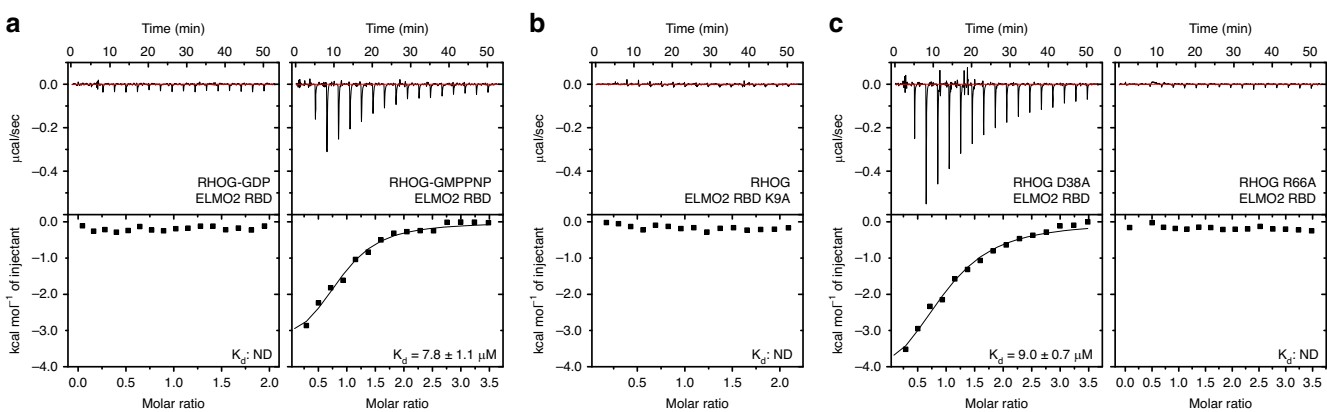

**Fig. 5 Mutations of the RHOG–ELMO2$^{RBD}$ interface disrupts interactions.** ITC binding assay for (**a**) Wild-type RHOG in the inactive GDP-bound state (left) or active GTP-bound state (right) with wild-type ELMO2$^{RBD}$; (**b**) K9A mutant of ELMO2$^{RBD}$ with wild-type RHOG; and (**c**) D38A and R66A mutants of activated RHOG with wild-type ELMO2$^{RBD}$. For ITC assays showing no binding, the $K_d$ was not determined (ND). The positions of these residues are shown in Fig. 4c. The experiments were performed independently three times with similar results.

this, a RHOG Asp38 to Ala mutant binds ELMO2$^{RBD}$ with an affinity of 9.0 μM, comparable to wild-type (Fig. 5c, left). A key RHOG side chain at the binding interface is Arg66 in the switch 2 region (Fig. 4c), and ITC analysis revealed a RHOG R66A mutant is unable to bind ELMO2$^{RBD}$ (Fig. 5c, right). Size exclusion chromatography and NMR verified that the two mutants that disrupt the RHOG-ELMO$^{RBD}$ interaction, RHOG$^{D38A}$ and ELMO$^{K9A}$, were correctly folded and comparable to wild type

(Supplementary Fig. 10). These data reveal that Arg66 of RHOG and Lys9 of ELMO2 are key residues driving formation of this complex, and corroborate the unique switch 1–2 recognition site of the ELMO ubiquitin-fold RBD (Supplementary Fig. 9b). While direct contact with the RAS switch 2 region is not typical of RAS-RBD interactions (Supplementary Fig. 9c–e), there are two RBD domains from RAS effectors that do contact switch 2 (via hydrophobic residues at the bottom of an auxiliary α-helix): RASSF5/NORE1A[55] and AF6/AFDN[56] (Supplementary Fig. 9f, g). Their binding mode, however, is entirely distinct from RHOG−ELMO2$^{RBD}$ which we suggest is a new class of GTPase-RBD complex.

**The closed conformation of DOCK2−ELMO1 is auto-inhibited**. The interaction of ELMO1$^{NTD}$ with DOCK2$^{DHR2}$ in the closed DOCK2−ELMO1 binary complex, such that ELMO1$^{RBD}$ occludes the RAC1-binding site of DOCK2$^{DHR2}$ suggested the possibility that ELMO1$^{NTD}$ suppresses DOCK2 activity. We therefore tested the effects of removing the steric hindrance caused by ELMO1$^{NTD}$ to RAC1 binding and GEF activity. When ELMO1$^{NTD}$ was deleted (DOCK2−ELMO1$^{\Delta NTD}$), the GEF activity increased by 60% (Fig. 6a, b and Table 1). This is consistent with the idea that the closed-conformation of ELMO1$^{NTD}$ represents an auto-inhibited state of DOCK2−ELMO1.

Superimposing the crystal structure of the ELMO2$^{RBD}$−RHOG complex onto ELMO1$^{RBD}$ of the DOCK2−ELMO1 complexes showed that for DOCK2−ELMO1−RAC1, with ELMO1$^{NTD}$ in the open-conformation, the RHOG-binding site on ELMO1$^{RBD}$ is readily accessible for RHOG engagement (Fig. 6c). However, the binding of RHOG to ELMO1$^{RBD}$ of the DOCK2−ELMO1 binary complex, with ELMO1$^{NTD}$ in the closed-conformation, is sterically occluded by DOCK2$^{DHR2}$ (Fig. 6d). Thus, RHOG binding to DOCK2−ELMO1 is incompatible with the closed-conformation. This suggests that RHOG engagement by ELMO1$^{RBD}$ might destabilize the closed conformation and promote the open-conformation, thereby relieving auto-inhibition. In accord with this suggestion, RHOG activates RAC1 by direct interaction with ELMO1$^{RBD}$ in vivo in the context of the plasma membrane[35].

We also compared the crystal structure of the ELMO2−BAI1 complex[42] with our DOCK2−ELMO1 complex (Fig. 6c, d). This revealed that in the closed-conformation of DOCK2−ELMO1, the binding site on ELMO1$^{EID}$ for the C-terminal helix of the GPCR BAI1 is sterically occluded by ELMO1$^{PH}$ (Fig. 6d), precluding BAI1 engagement to the auto-inhibited state of DOCK2−ELMO1. By contrast, in the open-conformation, the BAI1-binding site is accessible (Fig. 6c). Thus, similar to RHOG binding to ELMO1$^{RBD}$, engagement of BAI1 to ELMO1$^{EID}$ would promote the open, active conformation of DOCK2−ELMO1, stimulating its GEF activity. The mode of binding of BAI1 is conserved with BAI2 and BAI3 which interact with ELMO through a similar C-terminal helix[39].

The Tyro3, Axl, and Mer receptor tyrosine kinases (TAM kinases) phosphorylate Tyr18 and Tyr48 of ELMO1$^{RBD}$ to promote RAC activation and cell migration[57]. In the closed-conformation of the binary DOCK2−ELMO1 complex, Tyr18 is located at the ELMO1$^{RBD}$−DOCK2$^{DHR2}$ interface, whereas Tyr48 is solvent exposed (Fig. 4d). This suggests that phosphorylation of Tyr18 would destabilize the auto-inhibited conformation of DOCK2−ELMO1 in favour of the active state, consistent with the ability of TAM kinases to activate RAC.

**Phosphorylation of DOCK2 promotes RAC1 signalling**. Phosphorylation of DOCK1 on Tyr1811 is a possible mechanism for activation since it correlates with its activity toward RAC1[58,59].

While a similar site is not conserved within the equivalent C-terminal disordered region of DOCK2, the N-terminus of DOCK2 is instead phosphorylated on three proximal residues Tyr209, Tyr212 and Ser213 identified in multiple mass spectrometry studies (PhosphoSitePlus) (Fig. 7a). Our structure reveals that these sites are located within a phosphorylation linker segment (residues 208–218) connecting the helical region immediately C-terminal to DOCK2$^{SH3}$ with DOCK2$^{C2}$ (Fig. 1a, c). In the closed DOCK2−ELMO1 binary complex, this region, disordered in our structure, forms an interface with the ELMO1$^{ELMO}$ domain when the complex is auto-inhibited (Fig. 3e, left panel). As shown in Fig. 7a, alignment of different mammalian DOCK2 sequences surrounding these phosphorylation sites reveals that Tyr209, Tyr212 and Ser213 are conserved with the exception of X. tropicalis. These observations led us to reason that phosphorylation of DOCK2 at these sites may be a mechanism for the relief of the auto-inhibited state of the DOCK2−ELMO1 complex. To test this hypothesis, we generated a phospho-mimetic DOCK2 mutant where Tyr209, Tyr212 and Ser213 were mutated to Glu (DOCK2$^{YYS/EEE}$). We tested whether expressing a phospho-mimetic DOCK2 in cells would affect its GEF activity. By performing a GST-PAK pulldown assay to assess the levels of RAC activation, co-overexpressing DOCK2$^{YYS/EEE}$ with ELMO1 led to higher active RAC1 levels, compared with DOCK2$^{WT}$ (Fig. 7b), suggesting that DOCK2 phosphorylation on these sites can lead to increasing DOCK2 RAC1 GEF activity in HEK293T cells. We then reasoned that if DOCK2$^{YYS/EEE}$ can promote RAC1 GTP-loading, then it should enhance cell migration and invasion. Using Boyden migration and Matrigel-invasion assays, we found that co-expression of DOCK2$^{YYS/EEE}$ with ELMO1 and CRKII in HeLa cells led to higher cell migration and invasion compared with DOCK2$^{WT}$ (Fig. 7c, d). This was further confirmed in wound healing as well as time-lapse live imaging assays of cells co-expressing ELMO1 and CRKII with either DOCK2$^{YYS/EEE}$ or DOCK2$^{WT}$ (Fig. 7e, f, Supplementary Movie 2). CRKII was included because in functional assays, the RAC1-dependent activities of DOCK1 and DOCK2 are maximal when co-expressed with both ELMO1 and CRKII[12,30]. Collectively, these data reveal a mechanism whereby phosphorylation of DOCK2 in the phosphorylation linker likely alleviates DOCK2−ELMO1 auto-inhibition to catalyse RAC1 activation to enhance cell migration and invasion.

**ELMO1$^{NTD}$ is essential for RAC1 signalling by DOCK2 in cells**. In the auto-inhibited DOCK2−ELMO1 structure, the region of ELMO1$^{NTD}$, in particular ELMO1$^{RBD}$, occludes the DOCK2$^{DHR2}$ RAC1-binding site. Removal of ELMO1$^{NTD}$ increased the activity of the DOCK2−ELMO1 complex in vitro (Fig. 4a, b). We aimed to determine if removing ELMO1$^{NTD}$ would facilitate DOCK2-dependent RAC1 binding or activation in cells. We truncated the first 529 residues from ELMO1 to generate a mutant that lacks the N-terminal domain (ELMO1$^{\Delta NTD}$) (Fig. 8a). We tested whether the expression of ELMO1$^{\Delta NTD}$ in comparison to ELMO1$^{WT}$ has an effect on DOCK2 GEF activity by performing a GST-PAK pulldown assay to assess the levels of active RAC1. Co-expression of DOCK2 with ELMO1$^{WT}$ led to a significant increase in active RAC1 levels whereas co-expressing DOCK2 with ELMO1$^{\Delta NTD}$ led to a small but significant decrease in the levels of active RAC1 in comparison to the conditions with ELMO1$^{WT}$ (Fig. 8b). These results are consistent with a view suggesting that ELMO1$^{NTD}$ is required to target the DOCK2−ELMO1 complex to the membrane for efficient DOCK2-mediated RAC1 activation. To directly test if DOCK2$^{DHR2}$ is more accessible to bind RAC1 in the conditions where ELMO1$^{\Delta NTD}$ is co-expressed, we conducted nucleotide-

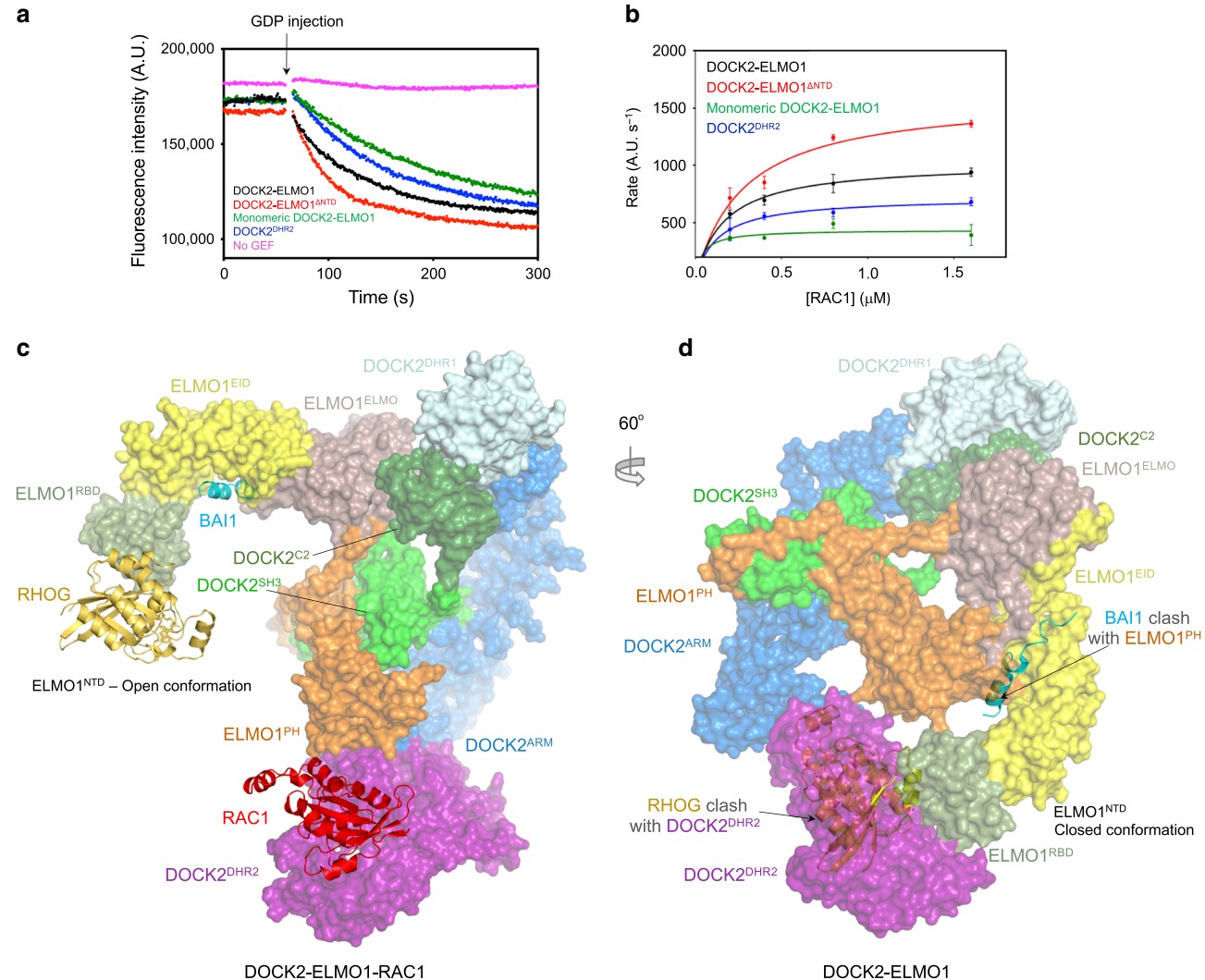

**Fig. 6 ELMO1$^{NTD}$ auto-inhibits DOCK2-ELMO1. a, b** GEF activity data. ELMO1 stimulates DOCK2 GEF activity. Deleting ELMO1$^{NTD}$ stimulates DOCK2–ELMO1 GEF activity 60%. Disrupting the DOCK2 dimerization reduces GEF activity by ~60%. **a** Experimental data for RAC1 at 0.8 μM. There was no observable spontaneous GDP exchange without the GEF. GEF activity data was monitored by the exchange of fluorescent mant-GTP following an injection of GDP after 1 min (indicated by arrow). **b** Rates as a function of RAC1 concentration. The initial rates of exchange at increasing substrate (mant-RAC) concentrations were fitted to a Michaelis-Menten equation with the resultant constants shown in Table 1. Data are presented as mean values with error bars of +/− one standard deviation. The experiment shown in (**a**) and (**b**) was performed six times. **c** The crystal structure of the RHOG–ELMO2$^{RBD}$ complex was superimposed onto the modelled ELMO1$^{RBD}$ of DOCK2–ELMO1-RAC1 (shown in surface representation with ELMO1$^{NTD}$ in the open-conformation). **d** The crystal structure of the RHOG–ELMO2$^{RBD}$ complex was superimposed onto ELMO1$^{RBD}$ of DOCK2–ELMO1 (shown in surface representation with ELMO1$^{NTD}$ in closed-conformation). This shows that RHOG-binding site on ELMO1$^{RBD}$ is exposed in DOCK2–ELMO1-RAC1, whereas in the binary DOCK2–ELMO complex, with ELMO1 in the down conformation, the RHOG-binding site on ELMO1$^{RBD}$ is occluded by DOCK2$^{DHR2}$. Modelling the ELMO2$^{NTD}$–BAI1 complex onto ELMO1$^{NTD}$ of the DOCK2–ELMO1 binary complex (closed-conformation) shows that BAI1 bound to the ELMO1$^{EID}$ domain would clash with ELMO1$^{PH}$ (**d**), but not in the ternary DOCK2–ELMO1-RAC1 complex (**c**). Thus, RhoG and BAI1 would only bind to the activated conformation of the binary DOCK2–ELMO1 complex with the ELMO1$^{NTD}$ in the open-conformation with the DOCK2$^{DHR2}$–RAC1 binding site exposed. Source data are provided as a Source Data file.

**Table 1 Initial rate constants of nucleotide exchange reactions of DOCK2-ELMO1 complexes.**

|  | DOCK2-ELMO1 | DOCK2-ELMO1$^{\Delta NTD}$ | Monomer: DOCK2-ELMO1 | DOCK2$^{DHR2}$ |
|---|---|---|---|---|
| Km (μM) | 0.17 | 0.28 | 0.04 | 0.13 |
| Vmax (arbitrary units. s$^{-1}$) | 1025 | 1604 | 438 | 720 |

free RAC1$^{G15A}$ pulldowns. GEFs form a stable complex with their target GTPases when in a nucleotide-free state[60]. Such an interaction is supported by structural evidence for DOCK2$^{DHR2}$-RAC1[44]. Hence, we generated the nucleotide-free RAC1$^{G15A}$ mutant, the equivalent of RHOA$^{G17A}$. We found that RAC1$^{G15A}$

binding to DOCK2 was minimally but significantly decreased upon expression of ELMO1$^{\Delta NTD}$ compared with conditions with ELMO1$^{WT}$ (Fig. 8c). Hence, this suggests that the decreased RAC1 activation upon expression of ELMO1$^{\Delta NTD}$ might be due to decreased levels of the DOCK2–RAC1 complex. Functionally,

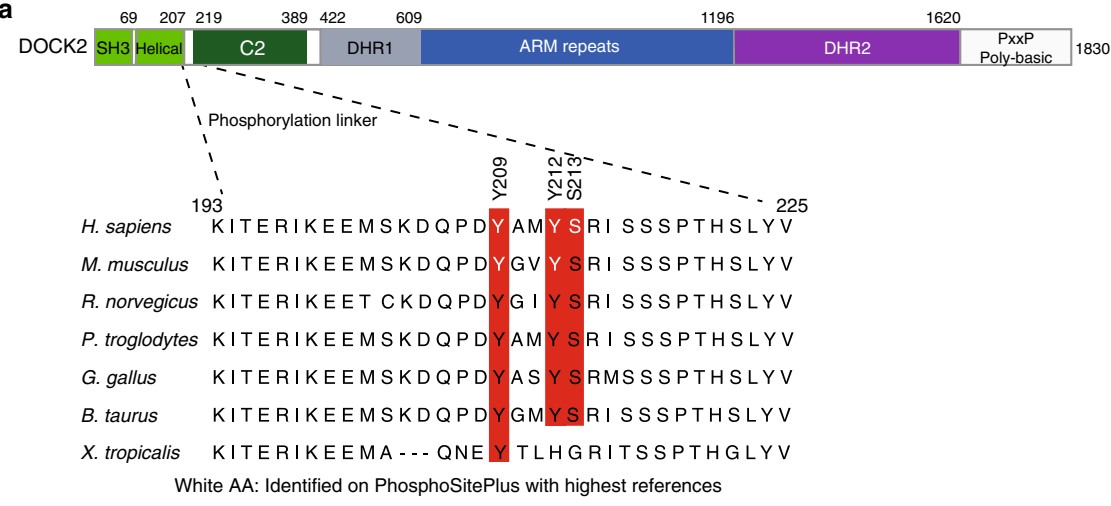

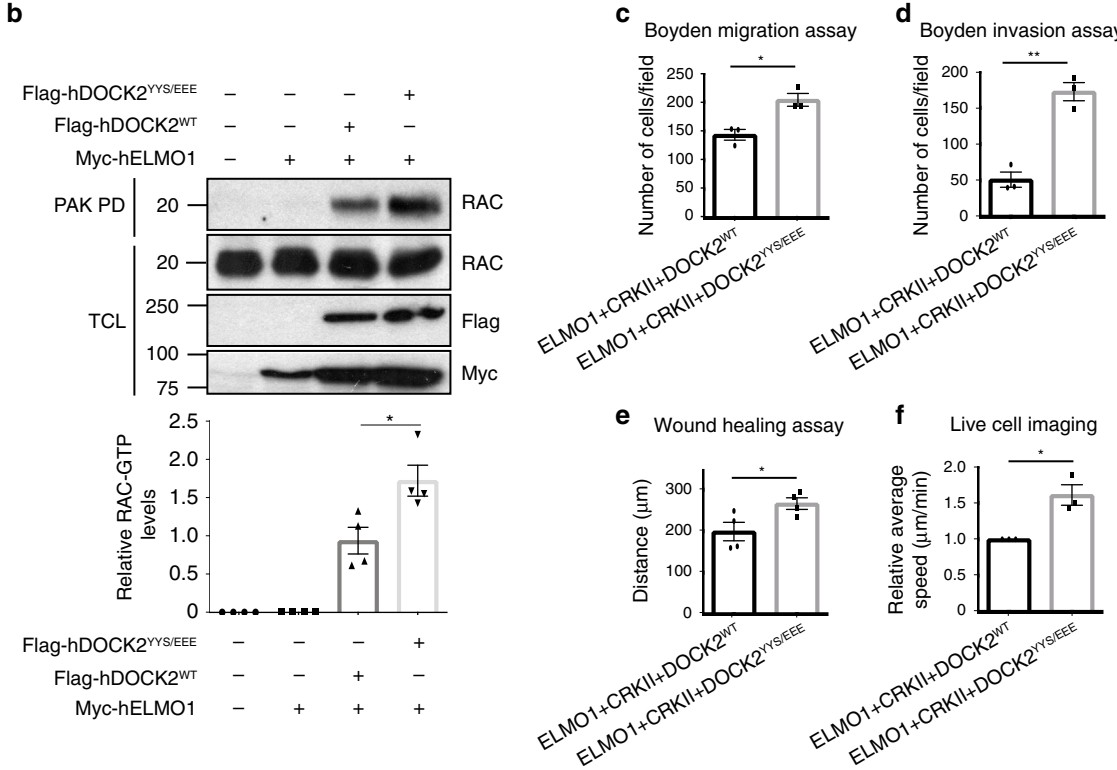

**Fig. 7 Phosphorylation of DOCK2 at the ELMO^NTD–DOCK2^C2 interface promotes RAC1 signalling. a** DOCK2 sequence alignment. Schematic of DOCK2 domains is shown. The aligned region is indicated within dashed lines. Sites highlighted in red are conserved within the different species. White labelled amino acids were identified on PhosphositePlus with the highest references. **b** RAC1 activation levels are increased upon DOCK2^YYS/EEE expression. 293 T cells transfected with the indicated plasmids were subjected to a RAC activation assay (GST-PAK1 PBD pulldown). RAC activation levels were detected by western blotting and quantified using ImageJ. Three biologically independent experiments were performed with similar results. Boyden migration (**c**) and invasion (**d**) assay performed with HeLa cells transfected with the indicated plasmids. Number of cells migrated were counted by DAPI staining of the membrane (the experiments were performed three times, imaging 10 fields using a 10X objective per condition). **e** Quantifications of transfected HeLa cells displacement in a wound healing assay. The experiments were performed four times, imaging four fields using a 10X objective per condition. **f** Average speed of transfected HeLa cells time-lapse imaged for 6 h. Average speed was quantified using manual tracking on ImageJ. The experiments were performed three times, imaging four fields used at 10X per condition per experiment. Data were analysed from three experiments and expressed as mean ± s.e.m. Two-tailed unpaired student's t test was used (**c, d, e, f**) and Mann–Whitney test (**b**). (*$p = 0.0286$ (**b**); *$p = 0.0138$ (**c**); **$p = 0.0017$ (**d**); *$p = 0.0424$ (**e**); *$p = 0.0123$ (**f**)). Source data are provided as a Source Data file.

co-expression of DOCK2 with ELMO1^ΔNTD and CRKII led to a decrease in the migration and invasion of HeLa cells, using Boyden migration and Matrigel-invasion assays, respectively (Fig. 8d, e). These results were further confirmed in wound healing and time-lapse live imaging assays, where cells expressing ELMO1^ΔNTD were less motile when compared with cells expressing ELMO1^WT (Fig. 8f, g and Supplementary Movie 2). Collectively, these data further confirm the importance of

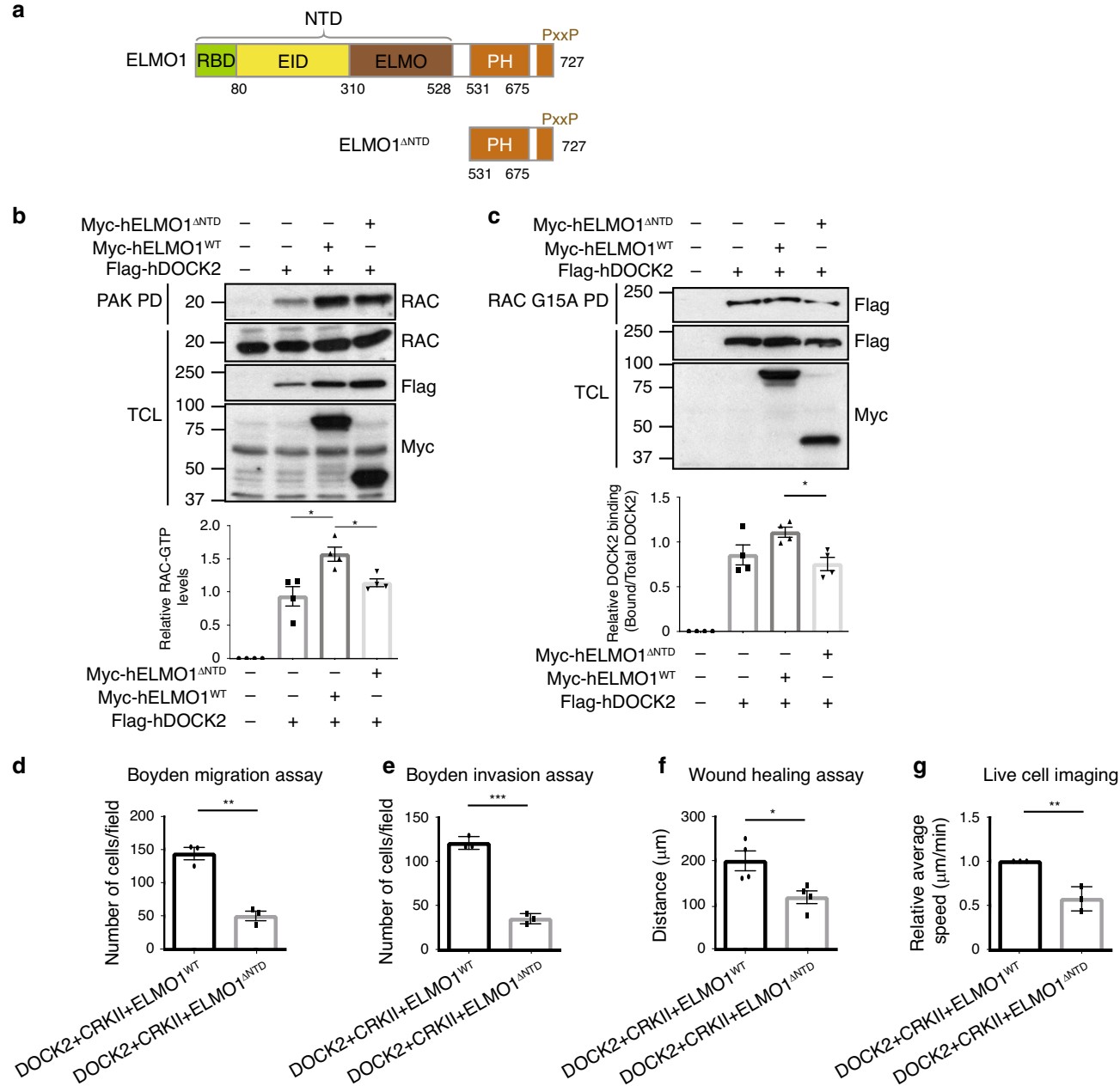

**Fig. 8 ELMO1^NTD is required for RAC1 signalling mediated by DOCK2. a** Schematic of full length ELMO1 and ELMO1^ΔNTD domains. **b** Expression of ELMO1^ΔNTD leads to less RAC1 activation when compared with ELMO1^WT expression. 293 T cells transfected with the indicated plasmids were subjected to a RAC activation assay (GST-PAK1 PBD pulldown). RAC activation levels were detected by western blotting and quantified using ImageJ. **c** DOCK2^WT binds less nucleotide-free RAC1 upon expression of ELMO1Δ^NTD. Lysates of 293 T cells expressing the indicated plasmids were used for GST-Rac1^G15A pulldown. Levels of Flag-DOCK2 bound to RAC1 were detected by western blotting and quantified by ImageJ. The experiments in (**b**) and (**c**) were performed four times. **d**, **e** Boyden migration (**d**) and invasion (**e**) assay performed with HeLa cells transfected with the indicated plasmids. Number of cells migrated were counted via DAPI staining of the membrane. (*n* = 3, 10 fields used at 10X per condition per experiment). **f** Quantifications of transfected HeLa cells displacement in a wound healing assay. Experiments were performed four times, imaging four fields using a 10X objective per condition. **g** Average speed of transfected HeLa cells time-lapse imaged for 6 h. Average speed was quantified using manual tracking on ImageJ. Experiments were performed four times, imaging four fields using a 10X objective per condition. Data were analysed from three experiments and expressed as mean ± s. e.m. Two-tailed unpaired student's *t*-test was used (**d**, **e**, **f**, **g**) and Mann–Whitney test (**b**, **c**). (*$p$ = 0.0286 (**b**, **c**); **$p$ = 0.0014 (**d**); ***$p$ = 0.000089 (**e**); *$p$ = 0.0216 (**f**); **$p$ = 0.0058 (**g**)). Source data are provided as a Source Data file.

ELMO1^NTD for optimal RAC1 activation and for the induction of RAC1-mediated cell migration and invasion in cells.

## Discussion
Our structure of DOCK2−ELMO1 reveals how the modular organization of two multiple domain subunits associate to form the binary complex. Conformational changes within ELMO1 relieves auto-inhibition suggesting a model for how interdependent ligand binding to multiple sites on both DOCK2 and ELMO1, and DOCK2−ELMO1 phosphorylation, regulates DOCK2 GEF activity (Fig. 9a). Contacts between domains within subunits create fairly rigid structures, with the only major conformational

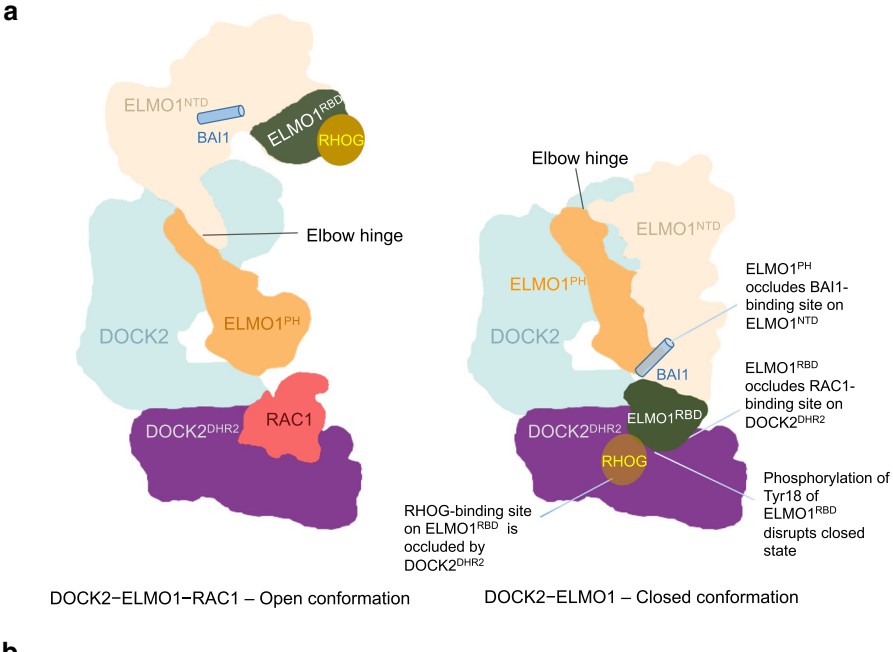

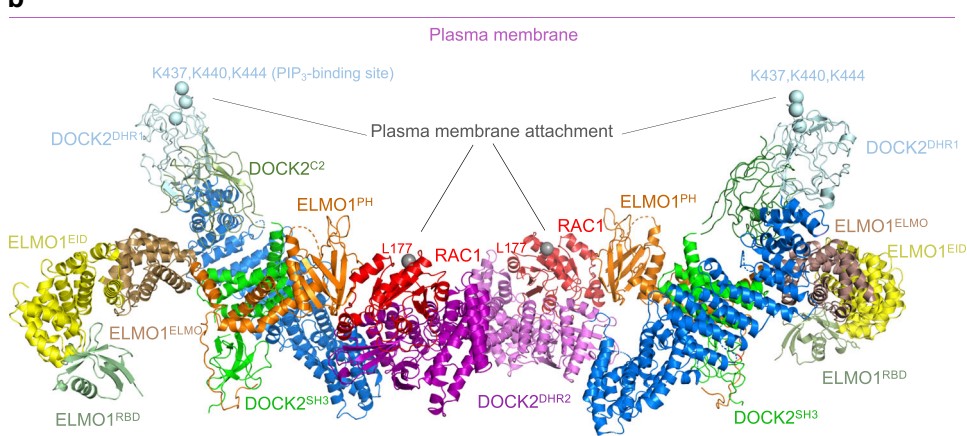

**Fig. 9 Schematic of conformational changes of DOCK2–ELMO1 and membrane attachment model. a** In the DOCK2–ELMO1–RAC1 ternary complex, ELMO1$^{NTD}$ adopts the open conformation with binding sites for RAC1, RHOG and BAI1 exposed on DOCK2$^{DHR2}$, ELMO1$^{RBD}$ and ELMO1$^{EID}$ (on ELMO1$^{NTD}$), respectively. In the closed, auto-inhibited state of the DOCK2–ELMO1 binary complex (the dominate state for DOCK2–ELMO1), rotation of ELMO1$^{NTD}$ by 120° about an elbow hinge connecting ELMO1$^{NTD}$ with ELMO1$^{PH}$ causes ELMO1$^{NTD}$ to form contacts with DOCK2$^{DHR2}$ and ELMO1$^{PH}$. These new interfaces create an auto-inhibited state that occludes binding sites for RAC1, RHOG and BAI. Phosphorylation of ELMO1 on either the phosphorylation linker (Fig. 1c) or Tyr 18 of ELMO1$^{RBD}$ (Fig. 4d) would disrupt the closed state. The view is similar to Figs. 3a, b and 6d. **b** Membrane attachment sites for DOCK2 and RAC1 are situated on the same face of the DOCK2–ELMO1–RAC1 complex. PIP$_3$ binds to a site defined by the L1 loop of DOCK2$^{DHR1}$ incorporating Lys437, Lys440 and Lys444, whereas RAC1 attaches to the membrane through a prenyl group attached to Cys189. Leu177 denotes the last ordered residue of the RAC1 crystal structure.

variability resulting from rotation about the hinge elbow connecting ELMO1$^{NTD}$ and ELMO1$^{PH}$. Interactions between the two subunits are centred on stable and invariant contacts between the N-terminal SH3 and helical domains of DOCK2 and the PH domain and poly-Pro segment at the C-terminus of ELMO1. The binary DOCK2–ELMO1 complex adopts two conformational states. In the closed, auto-inhibited conformation, ELMO1$^{NTD}$ is positioned to form extensive contacts with ELMO1$^{PH}$, DOCK2$^{C2}$ and DOCK2$^{DHR2}$ (Fig. 3e – right panel). Importantly, the latter interaction involving ELMO1$^{RBD}$ blocks access of DOCK2$^{DHR2}$ for RAC1, thereby suppressing DOCK2–ELMO1 GEF activity. In the alternative open, active conformation, ELMO1$^{NTD}$ has swung about the elbow hinge, exposing multiple binding sites on both DOCK2 and ELMO1 (Fig. 9). ELMO1 adopts the same open-conformation in the DOCK2–ELMO1–RAC1 ternary complex, indicating that the open-conformation of DOCK2–ELMO1

represents the active state of the complex. The bindings sites on DOCK2 and ELMO1 that become exposed in the open conformation comprise the RAC1-binding site on DOCK2$^{DHR2}$, the BAI-binding site on ELMO$^{EID}$, and the RHOG and Arl4A binding site on ELMO$^{RBD}$. Exposure of the ELMO$^{EID}$-binding site on BAI GPCRs upon receptor activation could then allow binding of ELMO and recruitment of DOCK–ELMO complexes to the plasma membrane[41]. Accessibility of the PIP$_3$-binding site on the DHR1 domain[23] of DOCK2 is unaffected by the conformational change of ELMO1$^{NTD}$ (Fig. 3e). Notably, the membrane attachment sites of the DOCK2–ELMO1 complex through PIP$_3$ engagement of DOCK2$^{DHR1}$, and RAC1, mediated by its C-terminal prenylation, are situated on the same face of the DOCK2–ELMO1–RAC1 ternary complex (Fig. 9b).

Activation of DOCK2 by relief of the ELMO1$^{RBD}$-mediated auto-inhibition requires a conformation change in the

ELMO1[NTD]. We propose that this could be regulated by two mechanisms. First, the binding of upstream regulators to distinct domains of ELMO, whose binding sites are blocked in the auto-inhibited state, for example RHOG or Arl4A to ELMO[RBD] and BAI receptors to ELMO[EID], would promote the active conformation. Second, phosphorylation of DOCK2 within its phosphorylation linker region, and ELMO1[RBD], whose sites are buried in the auto-inhibited state, would also promote the active conformation. Conceivably, prior phosphorylation of DOCK2−ELMO1 relieves auto-inhibition to expose binding sites for both BAI receptors and RHOG, recruiting DOCK2−ELMO1 to the cell membrane where RAC is localized. Because ELMO interacts with numerous additional proteins, such as the membrane protein ClipR-59, required for myoblast fusion[61], the model we propose here for relief of ELMO-mediated auto-inhibition may be general for numerous regulators of the DOCK−ELMO−RAC signalling module.

## Methods

**Cloning and mutagenesis.** The coding sequence for human *DOCK2* and *ELMO1* were amplified by PCR using primers 35–46 (see Supplementary Table 5 for a list of primers used in this study) and cloned into pF1 and pU1 plasmids, respectively[62]. A double StrepII tag together with a TEV cleavage site were attached to the C-terminus of DOCK2 and ELMO1. For RAC1 protein purification, pGEX-RAC1 was a gift from Jonathan Chernoff (Addgene plasmid # 12200). To disrupt DOCK2 dimerization, *DOCK2*[(Y1315A/L1322A/Y1329A)]-pF1 was prepared using USER methodology[63] using primers 31–34. ELMO1[ΔRBD] and ELMO[ΔNTD] were modified based on ELMO-pU1 by deletion of residues 2–79 and 2–529 with primers 27–30. Human *RHOG* (GeneID 391; amino acids 1–179) was cloned into the pDEST17 bacterial expression vector using Gateway technology, with a thrombin cleavage site inserted between the poly-HIS tag and the *RHOG* coding sequence using primers 15 and 16 (Supplementary Table 5). A sequence encoding the N-terminal RBD domain of murine *ELMO2* (GeneID 140579; amino acids 1–80) was cloned into a bacterial expression vector (pBR322) with an N-terminal glutathione S-transferase (GST) tag. To confirm the interaction interface of the RHOG and ELMO2 complex, RHOG[D38A], RHOG[R66A] and ELMO2[K9A] were prepared by the site directed mutagenesis method using primers 17–22. The plasmid pU1-*hELMO1* was used to generate pCS-6Myc-hELMO1 and pCS-6Myc-hELMO1[ΔNTD] by the Gateway cloning system using primers 1–4 (Supplementary Table 5). pCXN2-FLAG-hDOCK2 was a kind gift from Dr. Michiyuki Matsuda (Kyoto University) and was used to generate the *pCXN2-FLAG-hDOCK2*[YYS/EEE] mutant using the HiFi Gibson Assembly technology using primers 5–14 (Supplementary Table 5). pEGFP-C2-CRKII was a kind gift of Dr. Kristiina Vuori (Sandford Burnham Prebys Medical Discovery Institute).

**Expression and purification.** To express and purify the DOCK2−ELMO1 complexes, DOCK2−ELMO1, monomeric DOCK2−ELMO1, DOCK2−ELMO1[ΔNTD] and DOCK2−ELMO1[ΔRBD] complexes were co-expressed in High Five™ cells (BTI-TN-5B1-4) (ThermoFisher) for 2 days. The DOCK2−ELMO1 pellet was lysed in a buffer of 50 mM Tris HCl (pH 8.0), 200 mM NaCl, 2 mM DTT, 2 mM benzamidine, 1 mM EDTA, 0.2 mM PMSF and then loaded onto a Strep-Tactin® Column (QIAGEN) and the Strep-tagged complex was eluted with 5 mM desthiobiotin. The Strep-tag was cleaved by TEV protease overnight at 4 °C. The complex was further purified using Resource Q anion exchange chromatography and Superose 6 size exclusion chromatography. DOCK2[DHR2] and GST-RAC1 were expressed in *E. coli* B834 (DE3) cells (EMD Millipore) transformed with the pRare2 plasmid (Addgene) grown overnight at 20 °C. GST-RAC1 was purified using glutathione superflow (GE-Healthcare) followed by removal of the GST-tag using thrombin protease (Sigma). Cleaved RAC1 was further purified by Superose 6 size exclusion chromatography. The 5 mM EDTA was added to the buffer (50 mM Tris HCl (pH 8.0), 200 mM NaCl and 2 mM DTT) to remove nucleotide. DOCK2[DHR2] cell pellets were resuspended in five volumes of ice-cold lysis buffer (50 mM Tris HCl (pH 8.0), 350 mM NaCl, 2 mM imidazole, 2 mM β-mercaptoethanol, 10% (v/v) glycerol). His₆-Rac1 was purified by subjecting the lysate to Talon resin (Sigma Aldrich) and then to an S300 size exclusion column (GE Healthcare). To remove nucleotide, 5 mM EDTA was added to the buffer in the size exclusion purification step[43]. To prepare the DOCK2−ELMO1−RAC1 ternary complex, the DOCK2−ELMO1 binary complex and RAC1 were incubated in 1:2 molar ratio on ice in a buffer of 20 mM Hepes (pH 8.0), 200 mM NaCl for 30 min and crosslinked by 0.03% glutaraldehyde for 10 min on ice. The reaction was quenched by 50 mM Tris HCl (pH 8.0) and loaded onto a Superose 6 column for further purification. Size exclusion chromatography for the DOCK2−ELMO1 complex wild type and dimer mutant (Y1315A/L1322A/Y1329A: DOCK2[YLY/AAA]) was performed on a Superdex™ 200 5/150GL column, in gel filtration buffer 50 mM Tris HCl (pH 8.0), 200 mM NaCl and 2 mM DTT.

For RHOG purification, recombinant plasmids were transfected into *Escherichia coli* BL21 (DE3) CodonPlus cells (Agilent Technologies) and grown at 37 °C in Luria-Bertani medium with 100 µg/ml ampicillin up to optical density (at 600 nm) of 0.6. Protein expression was induced by the addition of 0.5 mM isopropyl-D-thiogalactopyranoside (IPTG) at 20 °C and cultures were grown for 18 h. Cells were lysed using sonication in 20 mM Tris HCl (pH 7.5), 150 mM NaCl, 5 mM MgCl₂, 10% (v/v) glycerol, 0.4% Nonidet P-40, protease inhibitors, and either 1 mM DTT or 5 mM β-mercaptoethanol. Lysates were cleared by centrifugation and incubated with nickel-nitrilotriacetic acid or GSH resin for 1–2 h at 4 °C. After washing in high-salt buffer (20 mM Tris HCl (pH 7.5), 500 mM NaCl, 5 mM MgCl₂, and 1 mM DTT or 5 mM β-mercaptoethanol), His-tagged proteins were eluted with 250 mM imidazole followed by thrombin cleavage. GST fusions were cleaved with thrombin directly on GSH resin overnight at 4 °C. For nucleotide exchange, RHOG proteins were incubated at 37 °C in the presence of 10 mM EDTA and a tenfold molar excess (nucleotide:protein) of GMPPNP (Sigma-Aldrich) for 10 min. 20 mM MgCl₂ was added and the sample was placed on ice. Cleaved and nucleotide-loaded proteins were purified on a HiLoad 26/600 Superdex 75 prep-grade column with a buffer consisting of 20 mM Tris HCl (pH 7.5), 100 mM NaCl, 5 mM MgCl₂, and 1 mM dithiothreitol. Fractions containing purified protein were pooled, concentrated to 10 mg/mL and stored at −80 °C prior to crystallization.

Single colonies of BL21 (DE3) CodonPlus bacteria (Agilent Technologies) transformed with GST-RAC1[G15A] or GST-PAK-PBD were grown in LB with antibiotics overnight at 37%. Then 24 h later, the bacteria cultures were grown in 5x the volume of LB with antibiotics and 0.1 M IPTG. These cultures were then shaken at 37 °C for 3 h. Bacterial pellets were obtained by centrifugation and suspended in 4 mL of lysis buffer (1xPBS, 1%Triton, 1x complete protein protease inhibitor (Roche)). After rupturing the cell membranes by sonication 3x of 30 s, lysates were cleared by centrifugation and supernatants containing the GST-proteins of interests were incubated with GST-beads for 1 h for affinity purification. The GST-beads were then washed and resuspended in 0.1% Triton-PBS.

**Fluorescence kinetics analysis.** The RAC1-mant GTP complex was prepared by incubating 2 µM RAC1 with 10 µM 2'/3'-O-(N'-methylanthraniloyl)guanosine 5'-O-triphosphate GTP (mant GTP) for 15 min at 20 °C in reaction buffer (50 mM Tris HCl (pH 8.0), 200 mM NaCl, 10 mM MgCl₂, 2 mM DTT)[44]. Fluorescence experiments were performed using a PHERAstar FS Plate reader (BMG LabTech). In the absence and presence of 0.2 µM GEF, varying concentrations of RAC1-mant GTP (from 0.2 to 1.6 µM) were incubated in reaction buffer (50 mM Tris HCl (pH 8.0), 200 mM NaCl, 10 mM MgCl₂, 2 mM DTT) for 10 min at 22 °C. The release reaction was initiated by injection of 100 µM GDP. The fluorescence signals (excitation wavelength 350 nm and emission wavelength 450 nm) were monitored every 0.6 s. The initial rates of exchange were fitted and the average from triplicate assays were plotted against concentration of substrate (RAC1-mant-GDP) and fitted to a Michaelis-Menten equation. Data were analysed using PRISM 8.2.1 (GraphPad Software). Graphs were plotted after subtraction of the uncatalysed nucleotide release rate.

**Electron microscopy.** Freshly purified DOCK2 samples (DOCK2−ELMO1−RAC1, DOCK2−ELMO1 or monomeric DOCK2−ELMO1−RAC1 mutant) were first visualized by negative-staining EM to check the sample quality. For cryo-EM grids preparation, DOCK2 samples were treated by 0.025% glutaraldehyde for 10 min on ice before a size exclusion chromatography purification using a Superose 6 Increase column (GE Healthcare) to remove aggregates. Without cross-linking treatment, most particles were found disassembled on cryo-EM grids. Aliquots of 3 µl samples at ~0.2 mg/ml were applied onto glow-discharged Quantifoil R1.2/1.3 holey carbon grids. The grids were incubated for 30 s at 4 °C and 100% humidity and then blotted for 8 s and plunged into liquid ethane using a Vitrobot III (Thermo Fisher).

DOCK2−ELMO1−RAC1 was imaged by a Thermo Fisher Scientific Titan Krios electron microscope at the MRC Laboratory of Molecular Biology (MRC-LMB) that was operated at an acceleration voltage of 300 kV and at a nominal magnification of 81,000 (resulting a calibrated physical pixel size of 1.43 Å/pixel), and a Gatan K2 summit detector that was installed after a GIF Quantum energy filter and operated with a slit width of 20 eV. In total, 1,338 micrographs were collected manually using K2 super-resolution mode. Each micrograph was exposed for 16 s at a dose rate of 5 electrons/pixel/sec and saved as 20 movie frames. Calculated defocus values are in a range of −1.5 to −3.2 µm. A second batch of 576 micrographs was collected on the same microscope at a nominal magnification of 64,000 (resulting calibrated pixel size of 1.76 Å/pixel).

DOCK2 − ELMO1 was imaged by two Titan Krios microscopes, one at the MRC-LMB and the other at eBIC, Diamond Light Source, operated at similar conditions to those for imaging DOCK2−ELMO1−RAC1 unless specified below. From MRC-LMB, 566 micrographs were collected manually using a K2 detector in counting mode at a nominal magnification of 64,000 (resulting calibrated pixel size of 1.76 Å/pixel). From eBIC, 4,025 micrographs were collected automatically by EPU using a K2 detector in counting mode (with calibrated pixel size of 1.34 Å/pixel).

**EM image processing**. All movie frames were aligned using MotionCor2[64] before subsequent processing. The contrast transfer function parameters were calculated using Gctf[65].

For DOCK2−ELMO1−RAC1, particles in 282 pixels x 282 pixels were selected by the automatic particle picking module in RELION 1.4, using reference-free 2D class averages from manually selected particles as templates[66–69]. The following steps were performed to exclude bad particles from the dataset. (1) Automatically picked particles in each micrograph were screened manually to remove ice contaminations; (2) Remaining particles were extracted and sorted by similarity to template images used for automatic picking, and those particles with low Z-scores were deleted; (3) 2-dimensional classification was performed and particles in bad classes with poorly recognizable features were excluded. 245,763 good particles were selected from two distinct datasets (Fig. 2a). For the second dataset collected at a lower magnification (1.76 Å/pixel), particles were rescaled to 1.43 Å/pixel when performing particle extraction in RELION[66–69]. Specifically, a box size of 230 pixels and a re-scaled box size of 282 pixels were used to scale the particles in Fourier space (230 pixels * 1.76 Å/pixel/282 pixels = 1.4354 Å/pixel).

To generate an initial model for 3D refinement, *e2initialmodel.py* program in EMAN2 package[45] was used with 10 selected 2D average images from RELION 2D classification. Initial refinement with a subset of 80,411 particles resulted in a map with 12~15 Å resolution. When a mask with ordered regions comprising the DOCK2$^{DHR2}$−RAC1 dimer, DOCK2$^{ARM}$, DOCK2$^{DHR1}$, DOCK2$^{C2}$ and DOCK2$^{SH3}$−ELMO1$^{PH}$ was used (Supplementary Fig. 1e), focussed refinement resulted in a significantly improved map at 6.6 Å resolution. By making a mask based on the improved map that only included strong densities, the refinement resulted in a 6.1 Å resolution map. Refinement using all particles resulted in a map at an overall resolution of 4.6 Å. The results indicated that flexibility was seriously influencing the alignment and two monomers needed to be separated to improve the resolution. We then prepared monomeric DOCK2−ELMO1−RAC1 complex by mutating the dimerization interface of DOCK2$^{DHR2}$ (Y1315A/L1322A/Y1329A). Although we could purify monomeric complex, the complex was very unstable on cryo-EM grids and we were not able to improve the resolution by this approach (Supplementary Fig. 6).

To improve the resolution, we separated two monomers computationally using signal subtraction[70]. With a mask that includes one monomer of DOCK2−ELMO1−RAC1 and the DOCK2$^{DHR2}$−RAC1 region from the other monomer, we performed a refinement to align all particles. We then subtracted the aligned monomer (all of the monomer but not DOCK2$^{DHR2}$−RAC1 region) from original particles to create a subtracted dataset A. With dataset A, we then performed a refinement with the mask of the other monomer map. We then used the alignment parameters to subtract the other monomer from the original dataset to create subtracted dataset B. By combining datasets A and B, a map was reconstructed to 4.1 Å resolution (Supplementary Fig. 2a, b).

From local resolution map and further 3D classification, we observed flexibility of individual parts/domains. For instance, DOCK2$^{ARM}$ and the DOCK2$^{SH3}$−ELMO1$^{PH}$ assembly can be distinguished into two conformations by 3D classification, respectively. To further improve the resolution of each part, we performed focussed refinement or focussed 3D classification in combination with refinement. A summary of all EM reconstructions obtained is listed in Supplementary Tables 1 and 2.

For DOCK2−ELMO1, data processing followed a similar procedure. The resolutions were generally lower compared with DOCK2−ELMO1−RAC1, likely due to the DOCK2−ELMO1 sample being less stable. However, the ELMO1$^{NTD}$ region was improved and showed more detailed structure.

**Map visualization**. Figures were generated using Pymol and Chimera[71].

**DOCK2−ELMO1 model building**. Initial model building for each domain was based on maps from DOCK2−ELMO1−RAC1, except for ELMO1$^{NTD}$ that was based on a map from DOCK2−ELMO1. For DOCK2$^{DHR2}$−RAC1, the crystal structure (PDB: 2YIN)[43] was fitted to the density map using "fit to map" program in Chimera[71] and then rebuilt in COOT[72] guided by side-chain densities (Supplementary Fig. 4a). For the ARM domain, 3D structure predictions from PHYRE2[73] and I-TASSER[74] were used to as guide for determining connections of helices, but the final map was built with poly-alanine. For the DHR1 domain, a structure model was firstly built using PHYRE2 based on the crystal structure of DOCK1$^{DHR1}$[23] and then fitted to the cryo-EM density map. For DOCK2$^{C2}$, a structure model was built ab initio using ROSETTA[47] and fitted into the density map. For the DOCK2$^{SH3}$−ELMO1$^{PH}$ assembly, the crystal structure of this region (PDB: 3A98)[31] was fitted as a rigid-body. The N-terminal extension of DOCK2 (a loop and α-helix) and ELMO1 (α-helix) beyond the crystal structure were built in COOT[72]. ELMO1$^{NTD}$ was built into the cryo-EM density based on a crystal structure of ELMO2 (PDB: 6IDX)[42]. All models were refined with PHENIX[75].

**RHOG−ELMO2$^{RBD}$ crystal structure determination**. Initial crystal screens for the RHOG and ELMO2$^{RBD}$ complex (1:1 molar ratio) were performed by sitting-drop vapour diffusion at 22 °C. After 2 days, microcrystals were obtained in 0.1 M CHES pH 9.5 and 1.0 M sodium citrate. These crystallization conditions were optimized

by the hanging-drop vapour diffusion method at 22 °C. Suitable crystals of native and selenium-methionine-derivatized versions were grown in 0.1 M CHES pH 8.8 and 0.95 M sodium citrate. For data collection, crystals were frozen in liquid nitrogen with 3.5 M sodium citrate as a cryoprotectant. X-ray data were collected at McGill Chemistry Characterization Facility of McGill University (Montreal, Canada) using a Bruker D8 Venture single crystal X-ray diffractometer, wavelength was 1.3417 Å, temp 100 K. Raw data were indexed, integrated, and scaled using Proteum software. Crystallographic statistics of data collection are provided in Supplementary Table 4.

The structure was solved by molecular replacement using Phaser in the PHENIX[75] software package. Model building and refinement were performed using COOT[72] and PHENIX[75]. The structure was validated with MolProbity. The statistics of structure refinement are provided in Supplementary Table 4. Coordinates and structure factors of the RHOG−ELMO2 RBD complex are deposited in the Protein Data Bank (PDB) with the accession code 6UKA. Ramachandran statistics: 97.9 allowed, 2.1 favoured.

**Isothermal titration calorimetry**. RHOG interactions with ELMO2 RBD domains were measured using a MicroCal ITC200 (Malvern). Stock solutions were diluted into filtered and degassed 20 mM Tris-HCl (pH 7.5), 100 mM NaCl and 1 mM DTT. Experiments were carried out at 25 °C. Wild-type or mutant ELMO2$^{RBD}$ were injected into a reaction cell containing RHOG wild-type or mutants. Fifty injections at 150 s intervals were performed. Data were fit using the Origin (version 7) software (OriginLab Corporation).

**Cell culture and transfections**. HEK293T (293T) and HeLa cells (both cell lines obtained from ATCC) were cultured in DMEM, supplemented with 10% foetal bovine serum (Gibco) and 1% Penicillin/streptomycin antibiotics (Wisent). HeLa cells were transfected using Lipofectamine 2000 (Invitrogen) according to manufacturer instructions. 293T cells were transfected using the calcium phosphate method.

**GST-pulldown assays and immunoblotting**. For the RAC activation assays, 293 T cells were lysed as described[59]. For GST-RAC1$^{G15A}$ pulldowns, 293T cells were lysed using 1% NP-40 (15 mM NaCl, 50 mM Tris pH7.5, 1% Nonidet P-40, 10 mM NaF, 1 mM Na$_4$P$_2$O$_7$, 1 mM Na$_3$VO$_4$, 1X complete protease inhibitor) buffer and cell lysates were cleared by centrifugation and incubated with the corresponding GST-tagged proteins on beads for 2 h at 4 °C (GST-RAC1$^{G15A}$ Pulldown) or 30 min at 4 °C (GST-PAK-PBD Pulldown). Lysates and GST-fusion bound complexes, were run on SDS-electrophoresis acrylamide gels at 180 V and transferred on nitrocellulose for 3 h at 4 °C at 50 V or overnight at 4 °C at 20 V. Immunoblots were then blocked with 1% BSA and incubated with the indicated primary antibodies overnight at 4 °C or room temperature. Immunoblots are then washed with 0.01% TBST three times and incubated with the corresponding secondary antibody for 30 min at room temperature. Protein signals were revealed via Clarity$^{TM}$ western ECL substrate (Biorad). Antibodies used: anti-Myc (Santa Cruz – SC40) dilution factor: 2,000; anti-RAC1 (EMD - Millipore 05389), dilution factor: 3,000; anti-FLAG (Sigma - A8592), dilution factor: 10,000. RAC activation and DOCK2–RAC1 binding levels were quantified by densitometry analysis using the ImageJ software program.

**Boyden migration and invasion assay**. Boyden assays were performed using 8 μm pores Boyden Chambers (24-well, Costar). For the invasion assays, the upper chamber was coated with 6 μL of Matrigel (BD Biosciences) dissolved in 100 μL of DMEM. HeLa cells were detached and washed with DMEM 0.1% BSA. In total, 100,000 cells were seeded in the top chamber and allowed to migrate for 6 h (migration) or 24 h (invasion) toward the bottom chamber containing 10% FBS. Upper and lower chambers were then washed with 1x PBS and cells on the bottom side of the chamber were fixed with 4% PFA. Cells in the upper chambers were removed using cotton swabs and the membrane was mounted on a glass slide using SlowFade Gold reagent (Invitrogen). The average number of migrating cells in 10 independent 20× microscope fields were evaluated, and each experiment was performed in triplicate.

**Time-lapse cell imaging**. HeLa cells plated on fibronectin-coated plates (1000 cells/well in a 12-well plate) were transfected with 1 μg Myc-hELMO1/Myc-hELMO1$^{ΔNTD}$, 3 μg Flag-DOCK2/Flag-DOCK2$^{YYS/EEE}$ and 0.5 μg GFP-CRKII. Then 48 h later, cells were imaged using Time-lapse microscopy at 10 min intervals for 6 h (Speed tracking) or 24 h (for wound healing), using phase contrast brightfield. Videos and images were obtained using Velocity and analysed via Image J software for their speed and distance measurements.

**Reporting summary**. Further information on research design is available in the Nature Research Reporting Summary linked to this article.

## Data availability

EM maps are deposited in the Electron Microscopy Data Bank under accession codes: 10498 (DOCK2−ELMO1−RAC1 ternary complex, open conformation), 10497 (DOCK2 − ELMO1 binary complex, closed conformation). Protein coordinates are deposited in the Protein Data Bank under accession codes: 6TGC (DOCK2−ELMO1−RAC1 ternary complex, open conformation), 6TGB (DOCK2−ELMO1−RAC1 ternary complex, open conformation) and 6UKA (ELMO2$^{RBD}$−RHOG). Plasmids and cell lines that were generated for and used in this study are available upon request from the authors. Source data are provided with this paper.

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

## Acknowledgements

This work was funded by the MRC Laboratory of Molecular Biology and MRC (MC_UP_1201/6) and Cancer Research UK (C576/A14109) grants to D.B. We are grateful to members of the Barford group for discussion; S. Chen, C. Savva and G. McMullan for help with the EM data collection; J. Grimmett and T. Darling for computing. This work was supported by grants from the Canadian Institutes for Health Research (CIHR) (to M.J.S. and J.F.C.) and the National Science and Engineering Council of Canada (NSERC) (M.J.S.). A.A.T and R.K. were supported by research studentships and fellowships, respectively, from the Fonds de recherche du Québec – Santé (FRQS). M.J.S. holds a Canada Research Chair in Cancer Signaling and Structural Biology. J.F.C. holds the TRANSAT Chair in Breast Cancer Research. We acknowledge Diamond Light Source for access and support of the cryo-EM facilities at the UK national electron bio-imaging centre (eBIC), proposal EM13708, funded by the Wellcome Trust, MRC and BBSRC.

## Author contributions

The project was conceived by D.B., A.A.T., J.F.C., M.J.S., Experiments were performed by D.B., L.C., J.Y., Z.Z., A.B., S.H.M., C.H.J., R.C.K., M.J.S., A.A.T. The paper was written by D.B., C.H.J., L.C., A.B., A.A.T., J.F.C., M.J.S.

## Competing interests

The authors declare no competing interests.
