## [Peer Review File · Nature Communications]

Reviewers' comments:

Reviewer #1 (Remarks to the Author):

The manuscript by Chang et al. reports the cryo-EM structure of the complex between the small GTPase Rac1, its guanine nucleotide exchange factor (GEF) DOCK2 and the DOCK2 activator ELMO1. The study also includes the cryo-EM structure of DOCK2-ELMO1 and the crystal structure of the RBD domain of ELMO1 bound to the small GTPase RhoG, the validation of several structural observations by mutagenesis and an analysis of potential regulatory phosphorylation sites in cells.

This work is an important and timely advance in the field of Rac GTPases, which has been lacking structures of full-length, fully active GEFs. Notably, no structure of full-length GEFs of the DOCK family, which are large multidomain GEFs that require other multidomain proteins such as ELMO1 to reach their active state, was previously available. DOCK2, whose structure is investigated in this study, has important functions in cytoskeletal rearrangements required for lymphocyte migration in response of chemokines, which are mediated by their activation of the small GTPase Rac by stimulation of GDP to GTP exchange. DOCK2 is also a GTPase regulator of biomedical importance, in which mutations cause a form of combined immunodeficiency.

The cryo-EM structures highlight many new and important features of the active complex: the overall organization of DOCK2 and ELMO1, an elaborate network of intramolecular and intermolecular interactions between DOCK2 and ELMO, and the existence of two dramatically different conformations, one that can readily bind Rac1 (coined "up"), and one in which ELMO1 forms autoinhibitory interactions with the Rac-binding site (coined "down"). Combined with the crystal structure of the RBD domain of ELMO bound to the small GTPase RhoG, an upstream regulator of DOCK2, the cryo-EM structures suggest that the "down" conformation is also unable to bind RhoG. The authors hypothesize that inhibition release may result from phosphorylation of a segment located between the SH3 and C2 domains of DOCK2. Consistent with this hypothesis, mutation of potential phosphorylation sites increases of Rac1-GTP level in cells in a weak although significant manner.

That being said, while the findings are exciting, the manuscript in its current form is rushed and difficult to read. The authors therefore fail to convey their message in a proper manner, which does not do justice to the quality and importance of their work. It is therefore necessary that the authors carefully revise their manuscript and figures to fix the numerous problems. Specific comments and a non-exhaustive list of important improvements in the text and figures are provided below.

Specific comments

Two different "down" conformations are shown, which seem to have significant differences. Can the authors clarify?

Figure 4a: Top panel: Provide a control experiment showing spontaneous dissociation

Figure 4a: Explain how the curve and table shown in the middle and bottom panels were determined and discuss them in the text.

The authors should discuss, at the light of their structural findings, why they think DOCK2-ELMO1 is a dimer.

The authors should also discuss the orientation of the membrane-binding elements (DHR1 domain and Rac1) with respect to the membrane in the context of the full-length, ELMO-bound dimer.

A gel filtration profile confirming that the monomeric DOCK2 mutant is a monomer should be

provided. The mutations should be given and shown in the structure in a supplementary panel

A reference to the study showing that the PH domain of ELMO is not involved in membrane attachment should be provided. A description of the canonical binding site of the PH domain of ELMO1 and a discussion of why it is not interacting with lipids should be added.

An obvious concern in the analysis of the RhoG/RBD complex is that it may represent a non specific crystallographic complex. Notably, it is surprising that rather mild mutations (removal of a side chain) suffice to disrupt the interaction entirely as measured by ITC . Simple additional experiments should be provided to consolidate this observation. These should include gel filtration analysis confirming that RhoG forms a stable complex with the RBD of ELMO1, and basic quality controls showing that the mutants are properly folded (eg, gel filtration profiles and circular dichroism and their comparisons to the wild type RDB)

Figure 6: Errors in the units should be corrected

Figure 6: The mutations used for the ITC study should be shown in a close-up view.

Figure 7: A close-up view of phosphorylation sites should be provided

Could the authors explain what is CRKII and its role in the reported experiment?

Last paragraph: What is the rationale of using the Rac1-G15A mutant to measure the interaction with DOCK2? Why not use wild-type Rac1? A reference describing the biochemical properties of this mutant should be provided.

It would be useful that the authors include a schematic representation at the end of the manuscript to recapitulates the important aspects of their models of inhibited and active ELMO/DOCK2. Likewise It would be useful to accompany the figures describing intramolecular and intermolecular interactions by schematic representations (eg, Figures 2 and 3).

Revisions of text and figures

All structures should be presented before they are compared to other structures. For example, DOCK2/ELMO1 is compared to DOCK2/ELMO1/RAC and then presented in the next paragraph; the ELMO1-RhoG complex is extensively discussed in the ELMO1/DOCK2 paragraph and then presented in a subsequent paragraph. This is not logical and it makes the manuscript very hard to read.

All figures panels must be called and be briefly described in the main text, Their content must match closely their description in the text (eg: Supplementary figures 9c,d,e,f, g are not called; supplementary Figure 6a is called with reference of the role of the PH domain but it is does not address any property of this domain)

The layout of figure panels should correspond to the order in which they are called in the main text. Currently, this is not the case and this also contributes to making the manuscript difficult to read

Many structural figures (both main and supplementary) are poorly designed and difficult to read. This requires a major improvement.

Notably, when discussing a specific structural aspect, large overall views in which the relevant information is very small should be replaced by close-up views focused on the observation that is being described (eg: Figures 2, 3 and 4).

The consistency between colours (eg, the RBD domain in green in the structure and grey in the scheme) and representations (eg map is coloured in the same way as the domains in figure 3a but not figure 3b) must be carefully checked

Colours should be easy to distinguish (eg light yellow is very difficult to read), the same colours should not be used for 2 domains (eg SH3 in DOCK2 and RBD in ELMO)

Overloading the figures with both cartoon and surface representations can be avoided when the color coding is sufficient to identify the domains. Figures 4 b,c,d are examples where close-up views without surface would improve the readability.

Reviewer #2 (Remarks to the Author):

This interesting study investigates regulation of the DOCK2-ELMO1 signaling module by determining cryo-EM structures with and without RAC1 as well as a crystal structure of the RAS binding domain (RBD) of ELMO1 in complex with active GMPPNP-bound RHOG. The cryo-EM structures define the overall architecture of DOCK2-ELMO1, including interactions within and between subunits as well as conformational changes consistent with autoregulation. Comparison of the binary and ternary complex suggests that the binary complex adopts an autoinhibited conformation in which the binding site for RHOG is sterically occluded while the binding sites for RHOG and BAI GPCRs are buried as are phosphorylation sites. The crystal structure reveals that, despite having a typical ubiquitin-like fold, the interface between the ELMO1 RBD and RHOG involves both switch regions of RHOG and thus differs from the canonical RBD-Ras binding modality. The structural observations are supported by in vitro and cell-based analyses of binding, RAC1 activation, migration, invasion and wound healing. Together, these and other observations provide interesting structural and molecular mechanistic insights into autoregulation of DOCK2 GEF activity through a hinged conformational change in ELMO2.

Overall, the experiments appear to be done correctly, appropriately analyzed and reasonably interpreted. Apart from a few minor issues noted below, the manuscript is well written and presents high-quality observations of general interest to the signaling and cytoskeletal dynamics communities.

Up and down are somewhat confusing descriptors of conformation that only make sense for a specified orientation. Is there a reason for not using more conventional descriptors such as open and closed, which do not depend on orientation?

p. 11, line 373-374, "a switch II region which specifies interaction with the guanine base" doesn't make sense as switch II is nowhere near the base. Probably interaction with the gamma-phosphate was meant.

The text refers to switch I and switch II (roman numerals) whereas the corresponding regions are labeled Switch 1 and Switch 2 in Figures 2 and 5. Consistent usage would improve clarity.

Reviewer #3 (Remarks to the Author):

This manuscript by Chang and co-workers is a structural study of several complexes involved in the regulation of RHO GTPases. The authors explore several structures of DOCK2 (a guanine nucleotide exchange factor) in combination with its regulator, ELMO, and two small G-proteins, RAC1 and RHOG. The studied complexes are: DOCK2-ELMO1-RAC1 ternary complex (CryoEM); DOCK2-ELMO1 binary complex (CryoEM); and ELMO2RBD-RHOG (crystallography).

The authors find a large conformational change in ELMO that can explain the regulation mechanism. They observe that ELMO1NTD appears in up and down conformations while interacting with DOCK2. The up configuration dominates in the DOCK2-ELMO1-RAC1 ternary complex, while the down one is seen in the majority of the DOCK2-ELMO1 complexes. The down conformation blocks the binding site for RHO GTPases, consequently this is considered the auto-inhibited complex, while the up conformation is the active GEF state. Transition from one state to the other could be triggered by phosphorylation of linker regions and/or binding of other proteins playing as upstream regulators.

The DOCK2 complexes are dimeric and flexible. In the cryoEM studies the authors use subtraction of one of the monomers and refine the other, and vice versa, so they can reconstruct one DOCK2-ELMO1-RAC1 or DOCK2-ELMO1 monomer using two monomers per particle. Additionally, since some regions of the cryoEM maps were still poorly resolved, the image processing culminates with focused 3D classification and refinement. This way, while the first 3D map was at moderate resolution, the final sub-refined data improves significantly, and they can build atomic models in 3D densities about 4 Å resolution for DOCK2-ELMO1-RAC1. In the DOCK2-ELMO1 binary complex the resolution is lower, around 6 Å, but it does show the ELMO1NTD domain with more details. The overall cryoEM strategy is well suited, and the final maps seem to have enough structural details to support the conclusions.

The main concern regards the description of the cryoEM data and the image processing. It is really messy, and I am afraid that this needs to be clarified in a full new version.

1.-One serious discrepancy is on the collected datasets. In "Materials and Methods" section, at "Electron microscopy" the authors say that datasets are:

DOCK2-ELMO1-RAC1 collected in one microscope with 1,338 micrographs at 1.43 Å/px

DOCK2-ELMO1 collected in two microscopes with 4,025 micrographs at 1.34 Å/px and 566 micrographs at 1.76 Å/px

However, in Supplementary Figure 2, where the image processing for DOCK2-ELMO1-RAC1 complex is outlined there are two datasets:

1338 micrographs at 1.43 Å/px and 576 micrographs at 1.76 Å/px

So, it is very difficult to reconcile the methods section with the description in Supplementary Figure 2. One straightforward possibility is just a mistake between DOCK2-ELMO1-RAC1 and DOCK2-ELMO1 in Methods, but this does not match neither. So, the authors need to clarify this issue and be sure that they have not mixed datasets for DOCK2-ELMO1-RAC1 and DOCK2-ELMO1 during the processing described in Supplementary figure 2 (I guess not, but ...). Also, the dataset at 1.76 Å/px has 566 or 576 micrographs?

By the way, both batches of micrographs in the same figure are named "Batch 1"

2.-Following with the use of two datasets with different pixel sizes (in the same Supplementary figure 2). Merging of data with different sampling is not trivial, but the authors do not explain how was it done. My guess is that they have gone through the recently developed "Estimate anisotropic magnification" tool implemented in Relion 3.1. If this is the case, how was it done? Did the authors run the CTFRefine and the Refine3D more than once with the two datasets together? If the data in Supplementary figure 2 is correct, the pixel size difference between the two datasets is around 20%, and this is really high. Maybe the merged datasets are the ones with 1.34 and 1.43 Å/px, this would make sense.

3.-Again, if the workflow in Supplementary figure 2 is right, it is remarkable that the selection of particles from the two datasets at the level of the first 3D refinement shows very similar numbers (133,062 and 112,701 particles) while the number of micrographs in one dataset is more than twofold of the other (1338 and 576 respectively). Why is this difference in the number of good particles if the samples have been prepared the same way?

4.-Along the manuscript the authors refer to Supplementary Table 1 (sometimes Table S1) as a summary of all the cryoEM reconstructions obtained after focusing the image processing in different regions of the complexes. I guess that this table would provide information about the number of particles used for each region after 3D classification. For instance, Supplementary figure 3 shows the 3D classification refinement strategy for region ELMO1NTD, and the final number of images, but this information is missing for the rest of cryoEM maps, and Supplementary table 1 just show a range of resolutions for all the maps, but there are no details.

5.-Lines 775-776.

Remaining particles we extracted and sorted by similarity to reference images using RELION. Which references?

6.-At the beginning of the results section, on lines 162-163, the authors state that "Purified DOCK2-ELMO1 forms a stable complex with nucleotide-free RAC1"

With no further comment on this. However, to my surprise, in the Methods section it is clear that all the DOCK2 samples were crosslinked with glutaraldehyde. It is important to state in the main text of Results that the samples are crosslinked, and that the complexes disassemble on the cryoEM grids without the crosslinking (as stated in lines 743-746).

After crosslinking, there is another step of purification by gel filtration to remove aggregates, and this is the sample that goes to cryoEM. I guess that the SDS-PAGE pictures shown in Supplementary figures 1 and 5 are of samples that went through the same process but without crosslinking, right?

Minor thing. Lines 793-794 when referring to the DOCK2 mutant to get the monomeric DOCK2-ELMO1-RAC1 complex the authors mention Supplementary figure 6. I guess it should be Supplementary figure 5.

We much appreciate their positive comments and constructive and thoughtful suggestions. In response to the reviewers' comments we have performed additional experiments and revised the text and figures. These revisions have improved the study. We hope that we have satisfactorily addressed the referees' concerns and questions. Reviewers' comments in black, our responses in blue.

Reviewer #1

We thank the reviewer for their many constructive comments.

The manuscript by Chang et al. reports the cryo-EM structure of the complex between the small GTPase Rac1, its guanine nucleotide exchange factor (GEF) DOCK2 and the DOCK2 activator ELMO1. The study also includes the cryo-EM structure of DOCK2-ELMO1 and the crystal structure of the RBD domain of ELMO1 bound to the small GTPase RhoG, the validation of several structural observations by mutagenesis and an analysis of potential regulatory phosphorylation sites in cells.

This work is an important and timely advance in the field of Rac GTPases, which has been lacking structures of full-length, fully active GEFs. Notably, no structure of full-length GEFs of the DOCK family, which are large multidomain GEFs that require other multidomain proteins such as ELMO1 to reach their active state, was previously available. DOCK2, whose structure is investigated in this study, has important functions in cytoskeletal rearrangements required for lymphocyte migration in response of chemokines, which are mediated by their activation of the small GTPase Rac by stimulation of GDP to GTP exchange. DOCK2 is also a GTPase regulator of biomedical importance, in which mutations cause a form of combined immunodeficiency.

The cryo-EM structures highlight many new and important features of the active complex: the overall organization of DOCK2 and ELMO1, an elaborate network of intramolecular and intermolecular interactions between DOCK2 and ELMO, and the existence of two dramatically different conformations, one that can readily bind Rac1 (coined "up"), and one in which ELMO1 forms autoinhibitory interactions with the Rac-binding site (coined "down"). Combined with the crystal structure of the RBD domain of ELMO bound to the small GTPase RhoG, an upstream regulator of DOCK2, the cryo-EM structures suggest that the "down" conformation is also unable to bind RhoG. The authors hypothesize that inhibition release may result from phosphorylation of a segment located between the SH3 and C2 domains of DOCK2. Consistent with this hypothesis, mutation of potential phosphorylation sites increases of Rac1-GTP level in cells in a weak although significant manner.

That being said, while the findings are exciting, the manuscript in its current form is rushed and difficult to read. The authors therefore fail to convey their message in a proper manner, which does not do justice to the quality and importance of their work. It is therefore necessary that the authors carefully revise their manuscript and figures to fix the numerous

problems. Specific comments and a non-exhaustive list of important improvements in the text and figures are provided below.

This is a valid point and we address the specific comments below. Briefly, we revised and re-ordered the text by ensuring that each of the three new complexes determined in this study (DOCK2-ELMO1-RAC1, DOCK2-ELMO1 and ELMO2^{RBD}-RHOG) are described chronologically, and comparisons between them made after each has been described. Most of the main sub-headings are the same. The section ‘DOCK2–ELMO1 structure shows conformational change of ELMO1’ has been split into two. The text from the previous version that requires discussion of the ELMO2^{RBD}-RHOG complex (that follows) has been moved to a new section ‘The closed conformation of DOCK2–ELMO1 is auto-inhibited’ that follows the discussion of the ELMO2^{RBD}-RHOG complex.

We revised and re-ordered the figures accordingly such that Fig. 4, 5 and 6 (original manuscript) are now **Fig. 6, 4 and 5**, respectively. **Figure 9** shows the schematic and indicates membrane-binding sites on DOCK2 and RAC1.

Specific comments

1. Two different “down” conformations are shown, which seem to have significant differences. Can the authors clarify?

On page 8 (lines 293-290) of the original manuscript we briefly compared the down (now defined as closed) conformations of the ternary and binary complexes. The main difference between the two is that in the ternary DOCK2-ELMO1-RAC1 complex closed state, ELMO1^{NTD} adopts flexible conformations, whereas in the binary DOCK2-ELMO1 complex, ELMO1^{NTD} is rigid mainly due to contacts with DOCK2^{DHR2}. We have modified the text to clarify this point (**page 8, lines 273-277 (line numbers refer to DOCK2-ver38.doc - unmarked file)**).

2. Figure 4a: Top panel: (now Fig 6a) Provide a control experiment showing spontaneous dissociation

Thank you for this suggestion. Now included (magenta curve) in the re-numbered **Fig. 6a** (formally Figure 4a top panel).

3. Figure 4a: (now Fig. 6a) Explain how the curve and table shown in the middle and bottom panels were determined and discuss them in the text.

Added to the **figure 6b legend** and Methods section (**page 28, lines 1085-1089 and page 17, lines 614-616**, respectively).

4. The authors should discuss, at the light of their structural findings, why they think DOCK2-ELMO1 is a dimer.

The evidence for a dimer is from the structure and the molecular weight. Other DOCK proteins revealed by structures of their DHR2 domains are dimers, eg DOCK9^{DHR2} (ref. ¹), DOCK10^{DHR2} (DB unpublished), and as we previously published for DOCK2^{DHR2} (ref. ²). We do not know why DOCK proteins are dimers. There are no clear functional explanations.

One possibility is to allow cooperativity between the two catalytic sites of the dimer. We do not have evidence for this. Now discussed **page 6, second paragraph**.

5. The authors should also discuss the orientation of the membrane-binding elements (DHR1 domain and Rac1) with respect to the membrane in the context of the full-length, ELMO-bound dimer.

Thank you for this suggestion. The membrane-binding sites of DOCK2^{DHR1} and the membrane attachment of RAC1 are positioned on the same side of the complex. Mentioned on **page 14-15) (lines 514-517)** and **Figure 9b**.

6. A gel filtration profile confirming that the monomeric DOCK2 mutant is a monomer should be provided. The mutations should be given and shown in the structure in a supplementary panel

A gel filtration profile is shown in **Supplementary Figure 5b**. The dimer disruption mutations are indicated in Methods 'Cloning and Mutagenesis' and in Supplementary Figure 5c. These mutations are modelled on similar mutation introduced into the dimerization interface of DOCK9¹. Thank you for this suggestion.

7. A reference to the study showing that the PH domain of ELMO is not involved in membrane attachment should be provided. A description of the canonical binding site of the PH domain of ELMO1 and a discussion of why it is not interacting with lipids should be added.

For a reference that the PH domain of ELMO does not bind membranes we cite Komander et al. (2008)³. In this paper we determined the structure of the ELMO1 PH domain. This study revealed structural differences with canonical PH domains that would be incompatible with binding of negatively charged phosphoinositide lipids. Two of the basic residues of canonical PH domains that bind PIs are substituted with Trp and Asp (conserved in all ELMO sequences). Furthermore, neither full-length ELMO1 nor the isolated ELMO1 PH domain were capable of specific binding to any phosphorylated PI in vitro, in lipid-coated beads pulldown experiments or phospholipid overlay assays. On **page 7 (lines 250-252)** we briefly explain how the EMLO1 PH domain differs from canonical PH domains.

8. An obvious concern in the analysis of the RhoG/RBD complex is that it may represent a non-specific crystallographic complex. Notably, it is surprising that rather mild mutations (removal of a side chain) suffice to disrupt the interaction entirely as measured by ITC. Simple additional experiments should be provided to consolidate this observation. These should include gel filtration analysis confirming that RhoG forms a stable complex with the RBD of ELMO1, and basic quality controls showing that the mutants are properly folded (eg, gel filtration profiles and circular dichroism and their comparisons to the wild type RDB)

We agree with the reviewer that care must be taken while interpreting crystallized protein complexes, and this is the reason we validated our structure using several mutant GTPase and RBD proteins. As mutations in the RHOG-ELMO interface were designed based on our structural data, the ITC binding assays performed using soluble proteins nicely corroborate the structure and demonstrate that the complex is not a crystallographic artefact. Further, single amino acid substitutions are sufficient to disrupt all previously studied GTPase-RBD

interactions. An R89L mutation in the RAF1 RBD completely prevents binding to HRAS⁴, at least 5 distinct single amino acid substitutions fully inhibit PI3K RBD binding to NRAS⁵, a Q2148E mutation in the RBD of PLC ϵ results in an 18-fold lower affinity for HRAS⁶, and a K283A mutant of the RASSF5 RBD weakens its affinity for HRAS 240-fold⁷. This also holds true for GTPase interactions with effectors having alternative binding domain folds. We do not see strong retention of the ELMO RBD-RHOG complex during gel filtration, but this is typical for protein complexes of this affinity (K_d of 7.8 μ M). To satisfy the reviewers request and provide evidence that ELMO RBD, RHOG and derived mutant proteins are well folded in solution we performed additional experiments using NMR spectroscopy. These data along with gel filtration profiles for each protein are detailed in the new **Supplementary Figure 10** and explained in the text (**page 10, lines 365-367**). The elution profiles demonstrate that RHOG-D38A, RHOG-R66A, and ELMO-K9A are all monomeric and elute at the same volume as their wild-type counterparts. ¹H-¹⁵N HSQC spectra of the isotopically labelled mutants show each protein is well folded in solution, evidenced by the well-dispersed peaks and overlay of each mutant with a corresponding wild-type spectrum. These new data confirm that the derived mutants used for ITC are well folded, stable monomers in solution and that these single amino acid substitutions are sufficient to disrupt complex formation.

9. Figure 6 (**now Figure 5**): Errors in the units should be corrected

We think the reviewer is referring to our notation $K_d = \text{n.m.}$, with n.m. indicating ‘not measurable’. This abbreviation was not previously indicated in the **Figure 5 legend**. This has been corrected and we now use the notation K_d : ND (for not determined).

10. Figure 6: The mutations used for the ITC study should be shown in a close-up view.

The ELMO and RHOG mutants used for the ITC binding studies are at the interface of the GTPase-RBD complex. The side chains of ELMO K9 and RHOG R66/D38 are all visible in **Figure 4c**, which depicts the residues in question in close-up as a ribbons diagram, and are discussed extensively in the Results sections “Crystal structure of RHOG–GMPPNP complexed with ELMO2^{RBD},” and “Mutations inhibit RHOG–ELMO binding”. Further, the position of these residues and their evolutionary conservation is displayed in **Supplementary Figure 8**. We have made the close-up view more evident by referencing **Fig. 4c** when discussing the ITC binding results (**Fig. 5**), highlighting the K9 (ELMO), D38 and R66 (RHOG) more distinctly and referred to these residues in the **Figure 5 legend**. RHOG Asp38 and Arg66 and ELMO2^{RBD} Lys9 are more clearly labelled in **Fig. 4c**.

11. Figure 7: A close-up view of phosphorylation sites should be provided

In the structure the phosphorylation sites indicated in **Fig. 7** are within a disordered region. This is explained **page 12, lines 431-433**.

12. Could the authors explain what is CRKII and its role in the reported experiment?

In addition to ELMO, CRK-family adapters have been shown to physically interact with DOCK proteins from worm, flies to mammals. In functional assays, the RAC1-dependent activity of DOCK proteins (e.g. DOCK1 and DOCK2) is maximal when co-expressed with ELMO and CRKII. We used these conditions to test the function of the ELMO/DOCK complex in cell migration and invasion assays (**Fig 7 c-f and Fig 8 d-g**). Relevant references

are ^{8,9}. We now state the reason for co-expressing with CRKII in the text (**page 13, lines 450-452**)).

13. Last paragraph: What is the rationale of using the Rac1-G15A mutant to measure the interaction with DOCK2? Why not use wild-type Rac1? A reference describing the biochemical properties of this mutant should be provided.

GEFs form a stable complex with their target GTPases when in a nucleotide-free state ⁵. Such an interaction is supported by clear structural evidence for DOCK2^{DHR2}-RAC1 ¹. Hence, we generated the nucleotide-free RAC1G15A mutant, the equivalent of RHOA G17A, to probe the accessibility of DOCK2^{DHR2} in the conditions described in **Fig 8c**. We now explain this (**bottom of page 13**).

14. It would be useful that the authors include a schematic representation at the end of the manuscript to recapitulate the important aspects of their models of inhibited and active ELMO/DOCK2. Likewise It would be useful to accompany the figures describing intramolecular and intermolecular interactions by schematic representations (eg, Figures 2 and 3).

The suggestion of a schematic to summarize our main conclusions is a very helpful suggestion. This is now included in **Figure 9a**. We are not sure schematic representations in Figures 2 and 3 would be of much value and there are space limitations that would mean the main figures would have a reduced size, making them less clear.

Revisions of text and figures

15. All structures should be presented before they are compared to other structures. For example, DOCK2/ELMO1 is compared to DOCK2/ELMO1/RAC and then presented in the next paragraph; the ELMO1-RhoG complex is extensively discussed in the ELMO1/DOCK2 paragraph and then presented in a subsequent paragraph. This is not logical and it makes the manuscript very hard to read.

Thank you for raising this. The manuscript has been reorganized to address this very valid point.

16. All figures panels must be called and be briefly described in the main text, Their content must match closely their description in the text (eg: Supplementary figures 9c,d,e,f, g are not called; supplementary Figure 6a is called with reference of the role of the PH domain but it is does not address any property of this domain)

These points are now addressed.

17. The layout of figure panels should correspond to the order in which they are called in the main text. Currently, this is not the case and this also contributes to making the manuscript difficult to read.

This point is now addressed.

18. Many structural figures (both main and supplementary) are poorly designed and difficult to read. This requires a major improvement.

We have revised and improved the figures, particularly Figs 2, 3, 6.

19. Notably, when discussing a specific structural aspect, large overall views in which the relevant information is very small should be replaced by close-up views focused on the observation that is being described (eg: Figures 2, 3 and 4).

We have increased the size of the figures and close-up views for Figs 2, 3 and 6 (formally 4).

20. The consistency between colours (eg, the RBD domain in green in the structure and grey in the scheme) and representations (eg map is coloured in the same way as the domains in figure 3a but not figure 3b) must be carefully checked
Corrected.

21. Colours should be easy to distinguish (eg light yellow is very difficult to read), the same colours should not be used for 2 domains (eg SH3 in DOCK2 and RBD in ELMO)

We darkened the yellow labels for clarity. We were running out of different colours for the many domains. We used different shades of green for SH3 (bright) and RBD (smudge), C2 (forest).

22. Overloading the figures with both cartoon and surface representations can be avoided when the color coding is sufficient to identify the domains. Figures 4 b,c,d are examples where close-up views without surface would improve the readability.

Figure 6 (formally Fig. 4) has been re-organized, with Fig. 4c-left removed. The figure is simplified. We no longer show both a cartoon and surface for the same domain/subunit. However, we retain surfaces for DOCK2-ELMO1 because this better illustrates the point we are trying to make, which is that in the closed state (**Fig. 6d**) neither RHOG or BAI1 (shown as cartoons) can bind to DOCK2-ELMO1, whereas in the open state (Fig. 6c) these binding sites are available (as for RAC1).

Reviewer #2

We thank the reviewer for helpful comments.

This interesting study investigates regulation of the DOCK2-ELMO1 signaling module by determining cryo-EM structures with and without RAC1 as well as a crystal structure of the RAS binding domain (RBD) of ELMO1 in complex with active GMPPNP-bound RHOG. The cryo-EM structures define the overall architecture of DOCK2-ELMO1, including interactions within and between subunits as well as conformational changes consistent with autoregulation. Comparison of the binary and ternary complex suggests that the binary complex adopts an autoinhibited conformation in which the binding site for RHOG is sterically occluded while the binding sites for RHOG and BAI GPCRs are buried as are phosphorylation sites. The crystal structure reveals that, despite having a typical ubiquitin-like fold, the interface between the ELMO1 RBD and RHOG involves both switch regions of RHOG and thus differs from the canonical RBD-Ras binding modality. The structural observations are supported by in vitro and cell-based analyses of binding, RAC1 activation, migration, invasion and wound healing.

Together, these and other observations provide interesting structural and molecular mechanistic insights into autoregulation of DOCK2 GEF activity through a hinged conformational change in ELMO2.

Overall, the experiments appear to be done correctly, appropriately analyzed and reasonably interpreted. Apart from a few minor issues noted below, the manuscript is well written and presents high-quality observations of general interest to the signaling and cytoskeletal dynamics communities.

1. Up and down are somewhat confusing descriptors of conformation that only make sense for a specified orientation. Is there a reason for not using more conventional descriptors such as open and closed, which do not depend on orientation?

This is a very helpful suggestion. The use of open and closed is much more descriptive than up and down. We now adopt the open and closed convention for the up and down conformational states.

2. p. 11, line 373-374, "a switch II region which specifies interaction with the guanine base" doesn't make sense as switch II is nowhere near the base. Probably interaction with the gamma-phosphate was meant.

Thank you for this. Agreed, changed accordingly, **page 9, lines 318-320**.

3. The text refers to switch I and switch II (roman numerals) whereas the corresponding regions are labeled Switch 1 and Switch 2 in Figures 2 and 5. Consistent usage would improve clarity.

Now amended to switch 1 and switch 2 throughout.

Reviewer #3

We thank the reviewer for their many constructive comments.

This manuscript by Chang and co-workers is a structural study of several complexes involved in the regulation of RHO GTPases. The authors explore several structures of DOCK2 (a guanine nucleotide exchange factor) in combination with its regulator, ELMO, and two small G-proteins, RAC1 and RHOG. The studied complexes are: DOCK2-ELMO1-RAC1 ternary complex (CryoEM); DOCK2-ELMO1 binary complex (CryoEM); and ELMO2RBD-RHOG (crystallography).

The authors find a large conformational change in ELMO that can explain the regulation mechanism. They observe that ELMO1NTD appears in up and down conformations while interacting with DOCK2. The up configuration dominates in the DOCK2-ELMO1-RAC1 ternary complex, while the down one is seen in the majority of the DOCK2-ELMO1 complexes. The down conformation blocks the binding site for RHO GTPases, consequently this is considered the auto-inhibited complex, while the up conformation is the active GEF state. Transition from one state to the other could be triggered by phosphorylation of linker regions and/or binding of other proteins playing as upstream regulators.

The DOCK2 complexes are dimeric and flexible. In the cryoEM studies the authors use subtraction of one of the monomers and refine the other, and vice versa, so they can reconstruct one DOCK2-ELMO1-RAC1 or DOCK2-ELMO1 monomer using two

monomers per particle. Additionally, since some regions of the cryoEM maps were still poorly resolved, the image processing culminates with focused 3D classification and refinement. This way, while the first 3D map was at moderate resolution, the final sub-refined data improves significantly, and they can build atomic models in 3D densities about 4 Å resolution for DOCK2-ELMO1-RAC1. In the DOCK2-ELMO1 binary complex the resolution is lower, around 6 Å, but it does show the ELMONTD domain with more details.

The overall cryoEM strategy is well suited, and the final maps seem to have enough structural details to support the conclusions. The main concern regards the description of the cryoEM data and the image processing. It is really messy, and I am afraid that this needs to be clarified in a full new version.

1.-One serious discrepancy is on the collected datasets. In "Materials and Methods" section, at "Electron microscopy" the authors say that datasets are:

DOCK2-ELMO1-RAC1 collected in one microscope with 1,338 micrographs at 1.43 Å/px
DOCK2-ELMO1 collected in two microscopes with 4,025 micrographs at 1.34 Å/px and 566 micrographs at 1.76 Å/px. However, in Supplementary Figure 2, where the image processing for DOCK2-ELMO1-RAC1 complex is outlined there are two datasets: 1338 micrographs at 1.43 Å/px and 576 micrographs at 1.76 Å/px. So, it is very difficult to reconcile the methods section with the description in Supplementary Figure 2. One straightforward possibility is just a mistake between DOCK2-ELMO1-RAC1 and DOCK2-ELMO1 in Methods, but this does not match neither. So, the authors need to clarify this issue and be sure that they have not mixed datasets for DOCK2-ELMO1-RAC1 and DOCK2-ELMO1 during the processing described in Supplementary figure 2 (I guess not, but ...). Also, the dataset at 1.76 Å/px has 566 or 576 micrographs?

By the way, both batches of micrographs in the same figure are named "Batch 1"

Re: In the "Methods" section, the second batch of data for DOCK2-ELMO1-RAC1 was omitted by a mistake. We have now added a sentence "A second batch of 576 micrographs was collected on the same microscope at a nominal magnification of 64,000 (resulting calibrated pixel size of 1.76 Å/pixel)". In **Supplementary Figure 2**, only DOCK2-ELMO1-RAC1 was discussed. The second "Batch 1" should be "Batch 2", which is fixed now. A complete summary of cryo-EM data statistics is included in **Supplementary Tables 1 and 2**. We thank the reviewer for pointing out the problem in our presentation of cryo-EM data.

2.-Following with the use of two datasets with different pixel sizes (in the same Supplementary figure 2). Merging of data with different sampling is not trivial, but the authors do not explain how was it done. My guess is that they have gone through the recently developed "Estimate anisotropic magnification" tool implemented in Relion 3.1. If this is the case, how was it done? Did the authors run the CTFRefine and the Refine3D more than once with the two datasets together? If the data in Supplementary figure 2 is correct, the pixel size difference between the two datasets is around 20%, and this is really high. Maybe the merged datasets are the ones with 1.34 and 1.43 Å/px, this would make sense.

Re: For merging the two datasets (batch 1: 1.43 Å/px; batch 2: 1.76 Å/px) of DOCK2-ELMO1-RAC1 as shown in **Supplementary Figure 2**, images of batch 2 dataset were rescaled to 1.43 Å/px when performing particle extraction in RELION. Specifically, we extracted particles in batch 1 with a box size of 282 pixels. When extracting particles in batch 2 using RELION, we selected a box size of 230 pixels, and scaled particles to 282 pixels (performed in Fourier space in RELION). This operation scaled the particles of batch

2 to a pixel size of 1.43 Å/pixel ((230 pixels * 1.76 Å/pixel)/282 pixels =1.4354 Å/pixel). We have added the procedure for merging two datasets in the Methods section of the revised manuscript.

3.-Again, if the workflow in Supplementary figure 2 is right, it is remarkable that the selection of particles from the two datasets at the level of the first 3D refinement shows very similar numbers (133,062 and 112,701 particles) while the number of micrographs in one dataset is more than twofold of the other (1338 and 576 respectively). Why is this difference in the number of good particles if the samples have been prepared the same way?

Re: One reason for the difference of particles per micrograph is that the two batches of data were collected at different magnification. Each micrograph in batch 2 is ~1.5 times of that of batch 1 ($1.76^2/1.34^2=1.51$) in area. Taken this into account, the average number of particles in batch 1 is 99 (133,062/1338), whereas the number of particles in similar area in batch 2 is 129 (112,701/576/1.51). This remaining difference might be caused by variance in different grids.

4.-Along the manuscript the authors refers to Supplementary Table 1 (sometimes Table S1) as a summary of all the cryoEM reconstructions obtained after focusing the image processing in different regions of the complexes. I guess that this table would provide information about the number of particles used for each region after 3D classification. For instance, Supplementary figure 3 shows the 3D classification refinement strategy for region ELMO1NTD, and the final number of images, but this information is missing for the rest of cryoEM maps, and Supplementary table 1 just show a range of resolutions for all the maps, but there are no details.

Re: We have changed “Table S1” to “**Supplementary Table 1**” throughout. Details such as final number of particles, accuracy of rotational and translational alignments, and B-factor used for sharpening for all maps have been included in **Supplementary Table 2** in the revised manuscript.

As required by the journal, we used the Nature Communications Template Table for Cryo-EM data and statistics for Supplementary Table 1, hence Supplementary Table 2 for the details of the final number of particles, accuracy of rotational and translational alignments.

5.-Lines 775-776.

Remaining particles we extracted and sorted by similarity to reference images using RELION.

Which references?

Re: We used 2D class averages from manually selected particles as templates/references for automatic particle picking in RELION¹⁰. The particles were sorted by similarity to the templates/references. We have clarified this in the revised manuscript.

6.-At the beginning of the results section, on lines 162-163, the authors state that "Purified DOCK2-ELMO1 forms a stable complex with nucleotide-free RAC1" With no further comment on this. However, to my surprise, in the Methods section it is clear that all the DOCK2 samples were crosslinked with glutaraldehyde. It is important to state in the main text of Results that the samples are crosslinked, and that the complexes disassemble on the cryoEM grids without the crosslinking (as stated in lines 743-746).

Re: We have modified the following sentence in the main text to clarify that samples were crossed linked for cryo-EM analysis. Details for crosslinking were maintained in the Methods section.

“Purified DOCK2–ELMO1 forms a stable complex with nucleotide-free RAC1 (**Supplementary Fig. 1a, b**), which was further stabilized by crosslinking to alleviate disassociation during grid preparation.”

After crosslinking, there is another step of purification by gel filtration to remove aggregates, and this is the sample that goes to cryoEM. I guess that the SDS-PAGE pictures shown in Supplementary figures 1 and 5 are of samples that went through the same process but without crosslinking, right?

Re: Correct. The SDS-PAGE pictures shown in **Supplementary Figures 1 and 5** are of samples without crosslinking. We added the details in the figure legends.

7. Minor thing. Lines 793-794 when referring to the DOCK2 mutant to get the monomeric DOCK2-ELMO1-RAC1 complex the authors mention Supplementary figure 6. I guess it should be Supplementary figure 5.

Re: This has been corrected. Thank you for pointing this out.

References

- 1 Yang, J., Zhang, Z., Roe, S. M., Marshall, C. J. & Barford, D. Activation of Rho GTPases by DOCK exchange factors is mediated by a nucleotide sensor. *Science* **325**, 1398-1402, (2009).
- 2 Kulkarni, K., Yang, J., Zhang, Z. & Barford, D. Multiple Factors Confer Specific Cdc42 and Rac Protein Activation by Dedicator of Cytokinesis (DOCK) Nucleotide Exchange Factors. *The Journal of biological chemistry* **286**, 25341-25351, (2011).
- 3 Komander, D. *et al.* An alpha-helical extension of the ELMO1 pleckstrin homology domain mediates direct interaction to DOCK180 and is critical in Rac signaling. *Molecular biology of the cell* **19**, 4837-4851 (2008).
- 4 Block, C., Janknecht, R., Herrmann, C., Nassar, N. & Wittinghofer, A. Quantitative structure-activity analysis correlating Ras/Raf interaction in vitro to Raf activation in vivo. *Nat. Struct. Mol. Biol.* **3**, 244–251 (1996).
- 5 Pacold, M. E. *et al.* Crystal structure and functional analysis of Ras binding to its effector phosphoinositide 3-kinase gamma. *Cell* **103**, 931–43 (2000).
- 6 Bunney, T. D. *et al.* Structural and mechanistic insights into ras association domains of phospholipase C epsilon. *Mol. Cell* **21**, 495–507 (2006).
- 7 Stieglitz, B. *et al.* Novel type of Ras effector interaction established between tumour suppressor NORE1A and Ras switch II. *EMBO J.* **27**, 1995–2005 (2008).
- 8 Brugnera, E. *et al.* Unconventional Rac-GEF activity is mediated through the Dock180-ELMO complex. *Nat Cell Biol* **4**, 574-582 (2002).
- 9 Guilluy, C., Dubash, A. D. & Garcia-Mata, R. Analysis of RhoA and Rho GEF activity in whole cells and the cell nucleus. *Nat Protoc* **6**, 2050-2060, doi:10.1038/nprot.2011.411 (2011).
- 10 Scheres, S. H. Semi-automated selection of cryo-EM particles in RELION-1.3. *Journal of structural biology* **189**, 114-122, (2015).

REVIEWERS' COMMENTS:

Reviewer #1 (Remarks to the Author):

The authors have satisfactorily addressed my comments. In particular, reordering of the text and figures and implementation of the new summary figure have greatly improved the manuscript which can now be recommended for publication, provided the remaining minor issues in the figures are addressed:

Figure 3f shows a PIP3 lipid head group bound to the PH domain of ELMO. This is in contradiction with the main text of the manuscript (lines 250-252) and point 7 in the rebuttal letter, in which the PH domain of ELMO is described as unable to bind phospholipids. Please clarify or remove from figure.

Figure 5 : Please correct Kd units to micromolar (instead of millimolar/mM)

Figure 9b : Please change « plasma attachment » to « plasma membrane attachment »

Reviewer #3 (Remarks to the Author):

The authors have now clarified the concerns related to the cryoEM datasets. Also they have answered other minor points. I recommend now publication of this work.

We are pleased we addressed nearly all the referees' concerns. We have now revised the manuscript addressing referee 1's comments and your editorial comments (our response to referee 1 in blue).

Reviewer #1 (Remarks to the Author):

The authors have satisfactorily addressed my comments. In particular, reordering of the text and figures and implementation of the new summary figure have greatly improved the manuscript which can now be recommended for publication, provided the remaining minor issues in the figures are addressed:

1. Figure 3f shows a PIP3 lipid head group bound to the PH domain of ELMO. This is in contradiction with the main text of the manuscript (lines 250-252) and point 7 in the rebuttal letter, in which the PH domain of ELMO is described as unable to bind phospholipids. Please clarify or remove from figure.

We have removed the PIP3 lipid head group from Figure 3f.

2. Figure 5 : Please correct Kd units to micromolar (instead of millimolar/mM)

Figure 5: Corrected mM to \$\mu\$ M.

3. Figure 9b : Please change « plasma attachment » to « plasma membrane attachment »

Figure 9b Changed as suggested.

Reviewer #3 (Remarks to the Author):

The authors have now clarified the concerns related to the cryoEM datasets. Also they have answered other minor points. I recommend now publication of this work.